# We Generate What You Need: Efficient Data Supplement via Model Prediction Discrepancy for Heterogeneous Federated Learning

## Abstract

One emerging approach to mitigating data heterogeneity in Federated Learning (FL) is to employ diffusion models to generate synthetic data for clients, thereby aligning local data distributions with the global distribution. Prior work has primarily focused on balance-oriented augmentation, which assumes a balanced global class distribution and thus generates samples of rare classes to rebalance each client's local dataset. However, in practice, global data distributions are often inherently imbalanced. For example, in weather forecasting, certain regions naturally experience more rainy days than sunny days, resulting in inherently imbalanced global training and testing data for those regions. Moreover, privacy and communication constraints in FL hinder the server's ability to accurately estimate the global distribution, rendering balance-oriented augmentation suboptimal. This raises a key, underexplored challenge: How can synthetic data be generated and selected to align local distributions with the true, yet unknown, global distribution? To address this challenge, we propose a novel framework, FedDPD. The key insight behind our approach is that a model's performance implicitly reflects the data distribution it has been trained on. Based on this observation, we use the performance discrepancy between the local and global models to identify the regions where each client's local dataset is lacking, and supplement corresponding synthetic samples for clients. Furthermore, we adapt the diffusion model for each client through a preference-optimization paradigm, enabling it to generate data that better aligns with the true global distribution, addressing the specific gaps in the client's local data. Notably, our approach incurs no additional computational overhead for clients. Extensive experiments on multiple benchmarks demonstrate that FedDPD outperforms state-of-the-art methods, achieving up to 3.82% improvement, regardless of the global distribution's balance.

## 1 Introduction

Federated Learning (FL) (McMahan et al., 2016) enables decentralized clients to collaboratively train models without sharing raw data. A key challenge in FL is data heterogeneity, where clients possess non-Independent and Identically Distributed (non-IID) data. This heterogeneity often leads to model drift, where local models diverge from the global model during training, amplifying discrepancies between clients and ultimately degrading the global model's performance. To mitigate this drift, existing methods constrain local updates or align local models with the global objective (Sahu et al., 2018; Karimireddy et al., 2019; Li et al., 2021; Acar et al., 2021; Luo et al., 2021; Zhang et al., 2022). However, these approaches introduce a trade-off between fitting local data and maintaining alignment with the global model, often compromising the ability of local models to learn client-specific knowledge, especially when data heterogeneity is high.

To tackle these challenges, recent efforts have explored the use of diffusion models to generate synthetic data for clients, aiming to shift their local data distributions to align more closely with the global distribution, thereby reducing model drift during local training. Compared to regularization-based methods, this paradigm mitigates data heterogeneity at the source, thus avoiding introducing the trade-off between fitting local data and aligning with the global model.

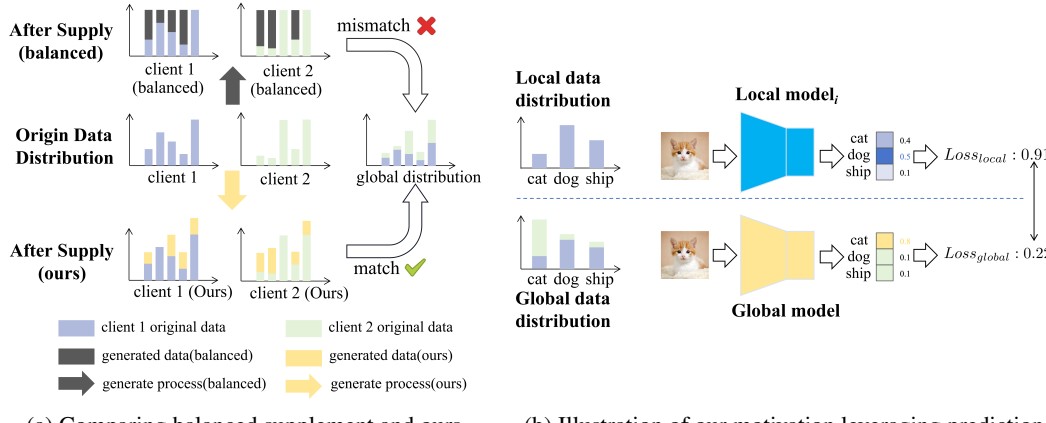

(a) Comparing balanced supplement and ours supplement.

(b) Illustration of our motivation leveraging prediction discrepancy.

Figure 1: (a) shows that balanced generation leads to a mismatch with the inherently imbalanced global distribution. (b) shows that we observe when a class is underrepresented in a client's local dataset, the local model incurs a higher loss compared to the global model.

We observe that existing methods for data supplementation in Federated Learning (FL), such as (Wen et al., 2022; Morafah et al., 2024; Qiang et al., 2025), commonly assume a *balanced global label distribution*, and accordingly generate synthetic samples to balance the class distribution within each client's local dataset. This assumption is reasonable in many critical domains, such as healthcare and autonomous driving, where the goal is to mitigate model bias and reduce false positives. In these scenarios, although individual clients may have imbalanced data, it is desirable for the global training and testing distributions to be balanced, so as to ensure fairness and minimize the risk of costly mispredictions.

However, in many real-world applications, *the global data distribution is inherently imbalanced*, and such imbalance is not only natural but also beneficial for model performance (Chou et al., 2021). For instance, in weather forecasting, some regions experience more rainy days than sunny days, and the global training datasets should reflect this imbalance to improve predictive accuracy. In such contexts, enforcing local class balance through synthetic data generation introduces a mismatch between training and testing distributions, which can degrade model performance. This challenge is further compounded in FL by strict privacy and communication constraints, which make it difficult for the server to accurately estimate the true global distribution. As a result, existing balance-oriented augmentation strategies become suboptimal in these settings. This raises a critical and underexplored question: **How can client-side data be synthesized and selected such that local distributions align with a true, yet unknown global distribution?**

To address the aforementioned challenge, we propose FedDPD, a novel framework that dynamically adjusts the data supplementation process according to the true global distribution, rather than assuming it to be balanced. The key insight behind FedDPD is that a model's behavior implicitly encodes the data distribution it has learned: when a model performs poorly on a given sample, it suggests that the sample lies outside the model's learned distribution.

Leveraging this insight, FedDPD quantifies the prediction gap between the global and local models on each generated sample, using this as a signal to guide selective supplementation. Specifically, the framework compares the losses incurred by the local and global models on each synthetic sample. If the local model yields a higher loss, it indicates that the client's local data distribution underrepresents this type of data. Such samples are then selectively allocated to the client for training, thereby implicitly aligning local and global distributions without requiring access to or revealing the client's private data. Figure 1a illustrates a toy example contrasting traditional balance-oriented augmentation with our proposed method. Figure 1b explains how prediction discrepancy between local and global models reveals underrepresented classes in a client's dataset.

Beyond selective data supplementation, we aim to customize the diffusion model to generate data tailored to the needs of each client. To instill a desired generative preference in the model, we adopt

a Direct Preference Optimization (DPO) paradigm (Rafailov et al., 2024) and train a lightweight LoRA module (Hu et al., 2021) for each client. We design a reward function, which encourages the diffusion model to generate samples with high client loss, low global loss, and disagreement between the client and global model predictions. This enables the generator to produce data that better captures what is lacking relative to the global distribution.

Importantly, FedDPD introduces no additional computational or communication overhead on the client side, making it both practical and efficient for real-world deployment. Furthermore, it is designed as a plug-and-play module that can be seamlessly integrated into standard FL pipelines, as well as heterogeneous FL settings involving diverse model architectures or resource-constrained clients.

We extensively evaluate FedDPD on multiple benchmark datasets and show that it consistently improves global model performance under non-IID conditions. Compared to state-of-the-art methods, FedDPD achieves higher accuracy and robustness, offering a scalable solution to data heterogeneity in FL.

## 2 RELATED WORK

### 2.1 DATA HETEROGENEITY IN FEDERATED LEARNING

Data heterogeneity, refers to the scenario where the data available to each client is not drawn from the same distribution, which is typical in real-world settings. This results that local models diverge from the global model as they are trained on different data distributions. As a result, the global model may struggle to generalize well across all clients, reducing its performance and effectiveness in practice. To handle non-IID data effectively, methods like FedProx (Sahu et al., 2018) introduce a regularization term that penalizes large deviations of local models from the global model, thereby constraining local updates. Similarly, SCAFFOLD (Sahu et al., 2018) uses control variates to reduce the variance in local updates, improving convergence in non-IID settings. More recently, MOON (Li et al., 2021) and FedAvgM (Hsu et al., 2019) have employed momentum and other strategies to alleviate drift. These methods mainly rely on aligning local models with the global objective rather than directly addressing the underlying data heterogeneity.

### 2.2 FEDERATED GENERATIVE LEARNING

Recently, several approaches in Federated Learning have been proposed to enhance model performance by using diffusion models to generate synthetic data. Gen-FedSD (Morafah et al., 2024) leverages Stable Diffusion to generate class-specific synthetic images on each client, with the goal of balancing label distribution and mitigating data heterogeneity in federated learning (FL). GenFL (Qiang et al., 2025) trains an auxiliary model solely on the generated data, and then aggregates it with the client models on the server side through weighted averaging. While these methods perform well when the global data distribution is balanced, they encounter significant challenges in scenarios where the global distribution is imbalanced, leading to mismatches between the client data distributions and the global distribution. Astraea (Duan et al., 2019) exposes the client data distributions to the server, which computes the categories that each client should supplement. While this method can align the local distributions with the global distribution in the case of an imbalanced global distribution, it reveals client label distributions to the server, which is impractical and raises significant privacy concerns.

Some approaches in one-shot Federated Learning also leverages synthetic data. Typically, these approaches (Zhang et al., 2024; Yang et al., 2024b;a; Chen et al., 2025) deploy a diffusion model on the client-side, where clients generate representative embeddings based on their local data. These embeddings are then uploaded to the server, which uses them to generate diverse synthetic images. However, directly transferring such methods into standard multi-round FL is non-trivial. To extract the representations in these methods, clients are typically required to run large-scale generative or vision-language models (e.g., diffusion models or BLIP-2) locally—an assumption that introduces substantial computational and memory overhead. While this may be acceptable in one-shot scenarios with a single round of communication, it becomes infeasible in regular FL settings, where the cost would be incurred repeatedly across rounds.

## 3 METHODOLOGY

### 3.1 PROBLEM DEFINITION

We consider a standard federated learning (FL) setting with $N$ clients $\mathcal{C} = \{1, 2, \ldots, N\}$. Each client $i \in \mathcal{C}$ owns a private dataset $\mathcal{D}_i = \{(x_j^{(i)}, y_j^{(i)})\}_{j=1}^{N_i}$ from its local data distribution $\mathcal{P}_i(x, y)$, which is not shared with other clients or the server. Let $\mathbf{d}_i = (d_{i,1}, d_{i,2}, \ldots, d_{i,C})$ denote the label distribution vector of client $i$, where $d_{i,c}$ denotes the proportion of samples from class $c$ in client $i$'s dataset. The global label distribution $\mathbf{d}_g = (d_{g,1}, d_{g,2}, \ldots, d_{g,C})$ is defined as:

$$d_{g,c} = \frac{\sum_{i=1}^N N_i \cdot d_{i,c}}{\sum_{i=1}^N N_i}, \quad c = 1, \ldots, C. \tag{1}$$

In our scenario, the local label distributions $\{\mathbf{d}_i\}_{i=1}^N$ are heterogeneous (non-IID across clients), and *the global label distribution $\mathbf{d}_g$ can also be inherently imbalanced.* The server aims to learn a global model $w_g$ that performs well under the unknown global label distribution $\mathbf{d}_g$. Formally, the objective can be written as maximizing the predictive accuracy:

$$w_g^* = \arg \max_w \mathbb{E}_{(x,y) \sim \mathcal{P}_g} \big[ \mathbf{1}\{h_w(x) = y\} \big], \tag{2}$$

where $\mathcal{P}_g(x, y)$ denotes the global data distribution with label prior $\mathbf{d}_g$, and $h_w : \mathcal{X} \to \mathcal{Y}$ is the prediction function parameterized by $w$.

### 3.2 PREDICTION DISCREPANCY GUIDED DATA SELECTION

Though the global distribution $\mathbf{d}_g$ is unknown in practice, we show that the prediction discrepancy between a local model $w_i$ and the global model $w_g$ on a given sample $(x, y)$ can reveal whether client $i$'s label distribution $\mathbf{d}_i$ is relatively deficient in class $y$ compared to $\mathbf{d}_g$.

Let $\ell(w; x, c)$ denotes the loss of model $w$ on the sample $(x, c)$. Following the Bayes' theorem (Bayes, 1763) and conclusion in (Gneiting & Raftery, 2007; Nguyen et al., 2010),we obtain this theorem(Concrete proof in Appendix A.2):

**Theorem 1.** *If $d_{i,c} < d_{g,c}$, then client $i$ suffers a strictly larger expected loss than the global model on class $c$:*

$$\mathbb{E}_{x \sim q(\cdot|c)} \big[ \ell(w_i; x, c) - \ell(w_g; x, c) \big] > 0. \tag{3}$$

**Basic Selection Mechanism** Building on the conclusion above, we design a practical mechanism to leverage prediction discrepancy for data supplementation in federated learning. In each communication round $t$ of FL, client $i$ locally trains a model $w_i^t$ on its private dataset $\mathcal{D}_i$ and uploads the updated model to the server. The server then aggregates all client models to obtain the global model $w_g^t$. We employs a text-to-image diffusion model on the server. In each round $t$, server randomly samples from a predefined prompt template set to generate image set $\hat{D}_i^t$. For each synthetic sample $(x_j, y_j) \in \hat{D}_i^t$, if:

$$\ell_i^t(x_j, y_j) - \ell_g^t(x_j, y_j) > 0 \tag{4}$$

it is assigned to client $i$ as supplemental data.

**Confidence-aware Filter** To prevent the supplementation of low-quality or uninformative samples, we introduce a confidence-aware filtering criterion. Specifically, we require that the global model not only achieves a lower loss on a sample but also exhibits sufficient prediction confidence. We define $\hat{p}_g^t(x_j)$ as:

$$\hat{p}_g^t(x_j) = \mathrm{softmax}(w_g^t(x_j))_{y_j}, \tag{5}$$

We require the confidence to exceed a random-guess probability $\tau = \frac{1}{C}$, where $C$ is the number of classes. The sample assignment rule is then updated as

$$(x_j, y_j) \in \tilde{\mathcal{D}}_i, \quad \text{if} \quad \ell_i^t(x_j, y_j) - \ell_g^t(x_j, y_j) > 0 \quad \text{and} \quad \hat{p}_g^t(x_j) > \tau \tag{6}$$

This additional constraint not only filters out low-quality samples but also stabilizes the optimization process: under cross-entropy loss, the gradient variance is bounded by $2(1 - \tau)$ (proof in Appendix A.3), thereby enhancing the stability of the supplementation process and preventing noisy gradients from destabilizing training.

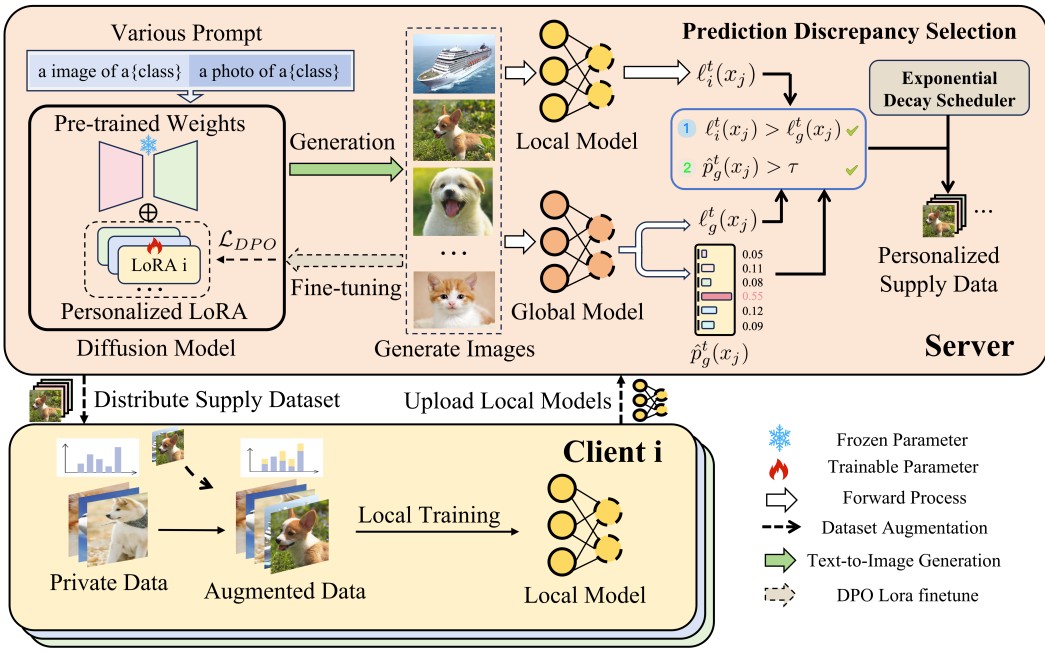

Figure 2: Overview of the FedDPD framework. The server generates synthetic images using a diffusion model and selects samples based on prediction discrepancy and global confidence, with the supply amount controlled by an Exponential Decay Scheduler. To further personalize generation, client-specific LoRA modules are finetuned via Discrepancy Preference Optimization (DPO). The selected synthetic samples are then distributed to clients, augmenting their private data and improving alignment with the global distribution.

**Exponential Decay scheduler** Although the prediction discrepancy mechanism helps align local distributions with the global distribution, we claim the quantity of synthetic data supplied in each round remains a critical factor in the training process. In the early stages of training, the model possesses limited knowledge, and the discrepancy signal may not accurately reflect the true distributional gap. On the other hand, supplying too little data early on may cause the model to converge to a poor local minimum, making it difficult to correct even with larger supplementation in later rounds. Inspired by various learning rate schedulers (Konar et al., 2020), we experiment with four representative forms: uniform, linear decay, stepwise decay, and exponential decay. The exponential strategy follows a simple normalized power-law decay:

Table 1: Accuracy comparison of different scheduling strategies on CIFAR-10 Dataset.

| Scheduler | $\alpha = 0.1$ | $\alpha = 0.5$ |
|---|---|---|
| Uniform | 52.07 | 51.35 |
| Linear | 53.75 | 55.14 |
| Stepwise | 55.56 | 55.98 |
| **Exponential** | **56.16** | **56.10** |

$$N_t = N_{\text{total}} \cdot \frac{1}{\sum_{s=1}^{T} s^{-\beta}} \cdot t^{-\beta}, \tag{7}$$

where $N_t$ is the number of samples supplied for each client, $N_{tot}$ is the fixed total number of samples supplied for each client, $\beta > 0$ is a hyperparameter controlling the decay rate. The detailed scheduling and training details is provided in Appendix A.4.

The empirical results are presented in Table 1, which demonstrate that the exponential decay scheduler achieves the best trade-off between early-stage acceleration and late-stage stability.

## 3.3 DISCREPANCY PREFERENCE OPTIMIZATION

Figure 3: Illustration of Discrepancy preference optimization module.

The selection mechanism described above effectively aligns each client's label distribution $\mathbf{d}_i$ with the global label distribution $\mathbf{d}_g$. Building on this foundation, we further adapt the diffusion model itself to enable personalized data generation, ensuring that each client receives synthetic samples tailored to its specific needs. Instead of acquiring entirely new knowledge, the diffusion model primarily needs to *adjust its generation bias toward the regions of the data space where each client is under-represented*.

To achieve this, we introduce the Discrepancy Preference Optimization paradigm, which leverages implicit preference signals to guide lightweight LoRA finetuning. Recall in each round $t$, the server uses diffusion model to generate synthetic images. For each sample $(x_j, y_j)$, we compute reward for each client $i$ as:

$$r_{i,j}^t = \hat{p}_i^t(x_j) - \hat{p}_g^t(x_j) \ + \ \lambda \cdot disagree(x_j), \tag{8}$$

where $\hat{p}_i^t(x_j)$ and $\hat{p}_g^t(x_j)$ are defined in Equation 5, and $disagree$ term is a binary indicator that equals 1 if and only if the global model and the local model predict different classes for $x_j$, and 0 otherwise.

$$disagree(x_j) = \mathbf{1}\{h_{w_i^t}(x_j) \neq h_{w_g^t}(x_j)\}. \tag{9}$$

The motivation behind this reward function is twofold. First, we encourage samples that are difficult for the client but easy for the global model, as this implies that such patterns are under-represented in the client's data distribution. Second, we encourage samples on which the client and global models make different predictions, since disagreement indicates regions of distributional mismatch where supplemental data can most effectively reduce divergence.

Based on the reward in Equation 8, we sort the generated samples for client $i$ in descending order. The top half of the $\hat{D}_i^t$ samples are regarded as *preferred* samples, while the bottom half are regarded as *rejected* samples. We then form preference pairs $\mathcal{S}_i^t = (x^+, x^-, y)$, where $x^+$ comes from the preferred set and $x^-$ from the rejected set with the same label $y$.

For each pair, let $\pi_\theta(x \mid y)$ denote the conditional likelihood under the current diffusion model with client-specific LoRA parameters $\theta$, and $\pi_{\text{ref}}(x \mid y)$ the reference model (a copy of pretrained diffusion model, without LoRA). Define the pairwise logit differences as

$$\Delta_\theta = \log \pi_\theta(x^+ \mid y) - \log \pi_\theta(x^- \mid y), \quad \Delta_{\text{ref}} = \log \pi_{\text{ref}}(x^+ \mid y) - \log \pi_{\text{ref}}(x^- \mid y). \tag{10}$$

Following the Direct Preference Optimization (DPO) paradigm, the training objective for client $i$ at round $t$ is

$$\mathcal{L}_{DPO}(\theta; i, t) = -\frac{1}{|\mathcal{S}_i^t|} \sum_{(x^+, x^-, y) \in \mathcal{S}_i^t} \log \sigma\Big(\kappa(\Delta_\theta - \Delta_{\text{ref}})\Big), \tag{11}$$

where $\sigma(\cdot)$ is the logistic function and $\kappa > 0$ controls the sharpness of preference alignment. The training details is shown in Appendix A.5

Table 2: Comparison of federated methods on CIFAR-10 and CIFAR-100 dataset.

| | CIFAR-10 | | | | CIFAR-100 | | |
|---|---|---|---|---|---|---|---|
| **Method** | $\alpha = 0.05$ | $\alpha = 0.1$ | $\alpha = 0.5$ | $\alpha = 1.0$ | $\alpha = 0.1$ | $\alpha = 0.5$ | $\alpha = 1.0$ |
| FedAvg (2016) | 51.58±2.46 | 51.36±3.82 | 50.60±2.89 | 52.98±0.99 | 34.19±1.18 | 29.63±2.02 | 27.73±2.21 |
| FedProx (2018) | 52.99±2.41 | 50.60±4.44 | 50.50±2.79 | 52.32±1.77 | 35.42±1.75 | 28.81±1.97 | 27.23±1.83 |
| FedProto (2022) | 52.18±2.25 | 50.38±4.09 | 50.10±2.92 | 54.02±1.54 | 34.87±1.32 | 29.39±1.68 | 27.61±1.30 |
| FedETF (2023) | 47.86±1.13 | 48.70±3.97 | 48.10±3.79 | 49.69±1.05 | 35.45±1.92 | 28.97±1.23 | 26.60±0.82 |
| FedFA (2023) | 45.43±2.16 | 47.67±3.42 | 50.24±1.98 | 52.16±1.58 | 32.37±3.08 | 28.46±1.61 | 26.21±1.98 |
| GenFL (2025) | 54.45±1.45 | 52.12±4.00 | 51.44±2.91 | 53.94±0.69 | 35.24±1.23 | 29.08±1.88 | 27.40±1.84 |
| Gen-FedSD (2024) | 63.36±1.80 | 58.30±3.73 | 53.27±2.70 | 56.20±1.52 | 36.80±1.60 | 31.09±1.04 | 29.68±1.77 |
| **FedDPD** | **64.64±2.23** | **59.96±3.67** | **57.09±3.14** | **58.95±1.37** | **37.41±1.12** | **32.44±1.75** | **31.58±1.45** |

Table 3: Comparison of federated methods on Tiny Imagenet dataset.

| | Tiny Imagenet | | |
|---|---|---|---|
| **Method** | $\alpha = 0.1$ | $\alpha = 0.5$ | $\alpha = 1.0$ |
| FedAvg (2016) | 20.94±0.80 | 15.41±1.06 | 15.61±0.33 |
| FedProx (2018) | 21.60±0.70 | 15.63±1.24 | 16.30±0.29 |
| FedProto (2022) | 22.07±0.60 | 15.68±0.83 | 15.88±0.48 |
| FedETF (2023) | 19.99±0.99 | 14.90±1.19 | 13.94±0.27 |
| FedFA (2023) | 21.21±0.44 | 13.49±1.10 | 13.45±0.47 |
| GenFL (2025) | 12.39±0.36 | 14.26±0.28 | 14.45±0.43 |
| Gen-FedSD (2024) | 13.88±0.11 | 15.57±0.17 | 15.96±0.41 |
| **FedDPD** | **23.27±0.37** | **17.56±0.42** | **17.75±0.43** |

Table 4: Average KL divergence for different data-supply strategies (lower is better).

| | CIFAR-10 | | | |
|---|---|---|---|---|
| **Method** | $\alpha = 0.05$ | $\alpha = 0.1$ | $\alpha = 0.5$ | $\alpha = 1.0$ |
| Origin | 1.376 | 1.416 | 0.656 | 0.359 |
| Random | 0.453 | 0.658 | 0.185 | 0.107 |
| Rare-class | 0.350 | 0.651 | 0.097 | 0.049 |
| **Ours** | **0.251** | **0.291** | **0.094** | **0.038** |

## 4 EXPERIMENT AND ANALYSIS

### 4.1 EXPERIMENTAL SETTINGS

#### 4.1.1 NETWORKS AND DATASETS.

Following prior studies, we evaluate the performance of our method and several state-of-the-art baselines on CIFAR-10 Krizhevsky et al. (2010), CIFAR-100 Krizhevsky (2009), and Tiny ImageNet Le & Yang (2015). We utilize Stable Diffusion v1-4 to generate synthetic data for clients and adopt ResNet He et al. (2016) as each client's local model. To simulate data scarcity and heterogeneity, we adopt the following configurations: 1. For CIFAR-10, we use the ResNet-8 model, each client having 500 samples. 2. For CIFAR-100, we use the ResNet-10 model, with 1000 samples per client. 3. For Tiny-ImageNet, we use the ResNet-10 model, with 1500 samples per client. In the above scenarios, the updates will be done across all 10 clients. To simulate globally imbalanced scenarios under different levels of heterogeneity, we modify the test data distribution to approximate the global label distribution.

#### 4.1.2 BASELINES AND IMPLEMENTATION.

We compare our proposed method against several representative baselines, including FedAvg, Fed-Prox, FedProto, FedETF, FedFA, GenFL, and Gen-FedSD. All methods are trained for 500 communication rounds using the SGD optimizer with a fixed learning rate of 0.1. For each baseline, we follow the original implementation and use the hyperparameters specified by the respective authors.

For Gen-FedSD, which generates balanced synthetic data for clients, we ensure that the total amount of synthetic data supplied per client matches that of our method. Specifically, in all scenarios, the quantity of generated data per client is set equal to the size of its initial local dataset. For Gen-FedSD, data supplementation is completed at the start of training. For our method, FedDPD, we set the supply coefficient $\beta$ to 1.0 for CIFAR-10 and Tiny-ImageNet datasets, and 1.25 for CIFAR-100. In the experiments on CIFAR-10, we select the Dirichlet coefficient $\alpha$ from {0.05, 0.1, 0.5, 1.0}. and {0.1, 0.5, 1.0} for CIFAR-100 and Tiny-Imagenet. For the above methods, we set each client to train for 5 local epochs before every aggregation round. For our method, FedDPD, we use a smaller number of local epochs, setting the local epoch to 1. For GenFL, Gen-FedSD and our

method FedDPD, we employ a variety of prompts to generate images. The text-to-image generation details is shown in Appendix A.6

## 4.2 CASE STUDY

In this section, we present a case study conducted using the CIFAR-10 dataset with $\alpha = 1.0$ under an imbalanced global distribution scenario. This case study is designed to demonstrate the effectiveness of our method in aligning local and global distributions from multiple perspectives.

Figure 4 illustrates the distributions from three different perspectives: (a) clients' data distribution before applying our framework, (b) clients' data distribution after applying our framework, and (c) the global data distribution.

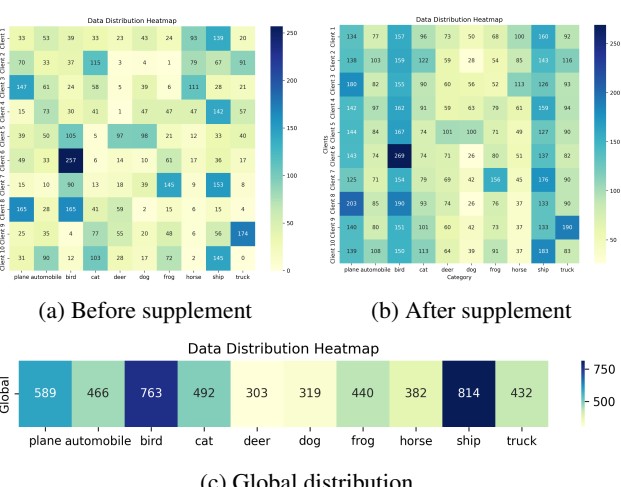

(a) Before supplement      (b) After supplement

(c) Global distribution

Figure 4: Data Distribution Heatmap

We can observe that the distribution in Figure 4b closely resembles that of Figure 4c. Then, we take Client 2 as an example. Under a balanced generative supplementation strategy, the algorithm would tend to prioritize the most underrepresented classes—such as classes 5, 6, and 7, which would result in a mismatch with the global distribution. In contrast, our framework instead supplements a large amount of data for class 2, which is also one of the most abundant classes in the global distribution. This demonstrates that our method effectively avoids misleading local supplementation and better aligns with the global data trend.

Beyond the qualitative case study, we further conduct a quantitative analysis on CIFAR-10 under different heterogeneity levels. Specifically, we evaluate the average KL divergence between each client's post-supplementation distribution and the true global distribution. We compare four strategies: the original distribution without supplementation, random supplementation, rare-class supplementation (prioritizing the most underrepresented classes, as in Gen-FedSD), and our proposed method. As shown in Table 4, our method consistently achieves the lowest KL divergence across all settings, substantially reducing the discrepancy between local and global distributions.

## 4.3 ANALYSIS

### 4.3.1 COMPUTATION AND PRIVACY ANALYSIS.

In this section, we discuss the computational consumption and privacy of our framework. Although our method supplements clients with additional synthetic data, we adopt only one local training epoch per communication round, in contrast to the five epochs used by other baselines. To quantify the effect, we calculate the total training FLOPs of ResNet-8 under different dataset sizes. As shown in Table 5, our method consistently requires less computation compared to FedAvg, acheving about 39.8% of the baseline cost.

On the server side, we integrate an additional text-to-image generation model. The actual generation of a single image can be completed within 2 second on an A100 GPU, demonstrating that our framework is computationally feasible for real-world applications.

In terms of privacy, our framework follows the standard federated learning setting where raw data never leaves the clients. Importantly, clients are not required to share their local label distributions with the server, which are often highly sensitive and could reveal private user information.

Table 5: Quantitative computational analysis for clients(TFLOPS)

|        | 500 samples | 1000 samples | 1500 samples |
|--------|-------------|--------------|--------------|
| Fedavg | 111.6       | 222.3        | 334.8        |
| Ours   | 44.5        | 88.4         | 132.9        |
| Ratio  | 0.3993      | 0.3977       | 0.3972       |

Table 6: Performance of combining our methods with FedProx and FedFA

| Method   | $\alpha = 0.1$ | $\alpha = 0.5$ | $\alpha = 1.0$ |
|----------|----------------|----------------|----------------|
| FedProx  | 34.17          | 29.89          | 27.93          |
| +FedDPD  | 36.38          | 34.65          | 32.00          |
| FedFA    | 37.61          | 35.82          | 33.16          |
| +FedDPD  | 39.39          | 36.07          | 35.20          |

### 4.3.2 PLUG-AND-PLAY FRAMEWORK.

In this section, we explore the potential of combining our proposed FedDPD framework with traditional FL methods. We take FedProx, FedFA and FedETF as examples. We conduct experiments combining FedDPD with FedProx, FedFA on the CIFAR-100 dataset with 10 clients, each having 1000 samples. We combine FedDPD with FedETF on the CIFAR-10 dataset, 10 clients having 500 samples. The experimental results with different dirichlet coefficient $\alpha$ is displayed in Table 6 and Table 7, which demonstrate a improvement over the single method.

Table 7: Performance of combining our methods with FedETF

| Method  | $\alpha = 0.05$ | $\alpha = 0.1$ | $\alpha = 0.5$ | $\alpha = 1.0$ |
|---------|-----------------|----------------|----------------|----------------|
| FedETF  | 51.11           | 44.49          | 46.48          | 53.48          |
| +FedDPD | 58.67           | 52.37          | 56.94          | 59.49          |

Table 8: Ablation Study

| PDS | EXS | DPPO | $\alpha = 0.1$ | $\alpha = 0.5$ | $\alpha = 1.0$ |
|-----|-----|------|----------------|----------------|----------------|
|     |     |      | 46.24          | 48.46          | 51.92          |
| ✓   |     |      | 52.07          | 51.35          | 55.82          |
| ✓   | ✓   |      | 54.35          | 54.53          | 56.79          |
|     |     | ✓    | 49.00          | 52.49          | 55.04          |
| ✓   | ✓   | ✓    | 55.44          | 54.90          | 57.03          |

## 4.4 ABLATION STUDY

To further investigate the contributions of different components of our FedDPD framework, we perform an ablation study on the CIFAR-10 dataset. In this study, we evaluate the impact of three key components of our method: the Prediction Discrepancy Selection (PDS), the Exponential supply Scheduler (EXS) and Discrepancy Direct Preference Optimization(DPPO). The experimental results is displayed in Table 8. For FedDPD (with PDS), we provide a default configuration where each client receives same number of random samples per communication round(as same as uniform scheduler). For FedDPD (with PDS and EXS), we follow the EXS formula to determine the number of selected samples to be added each round. For FedDPD(with DPPO), we have apply no selection strategy or scheduler. Result shows that these three modules both contribute for the improvement of global model.

## 4.5 DIFFERENT GENERATIVE MODEL ANALYSIS

We further conduct experiments on the CIFAR-100 dataset to evaluate the robustness of our framework across different generative backbones.

As shown in Table 9 in Appendix, our method consistently achieves performance improvements under both generative models, demonstrating the robustness of the proposed framework with respect to the choice of generative backbone.

## 5 CONCLUSION

We introduce FedDPD, a synthetic-data framework that leverages prediction discrepancy to align local and global distributions—without assuming global balance—and thereby improves robustness under class imbalance and non-IID settings on CIFAR-10, CIFAR-100 and Tiny-ImageNet. Future work will assess its scalability to domain-specific and domain-shifted datasets and its integration as a plug-in for existing federated-learning methods.

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

# A    APPENDIX

## A.1    PSEUDO CODE

**Algorithm 1** Federated Data Distribution Alignment Framework

**Input**: pretrained diffusion model $\theta$, local dataset $D_i$, number of rounds $T$, Exponential decay rate $\beta$, confidence threshold $\tau = \frac{1}{C}$, maximum supply data amount $D_{\text{total}}$ for each client

            **Output**: Updated global model $w_g^T$

1: Initialize global model $w_g^0$, local models $w_i^0$ for each client $i$
2: Set $t = 0$

3: **while** $t \leq T$ **do**
4:    `Client executes:`
5:    Train local model: `LocalTrain`$(w_i^t, D_i)$
6:    Upload local models $w_i^t$ to the server
7:    `Server executes:`
8:    Server aggregates local models to get global model $w_g^t$
9:    Get Supply Amount: $N_t = N_{\text{total}} \cdot \frac{1}{\sum_{t=1}^T t^{-\beta}} \cdot t^{-\beta}$
10:    **for** each client $i$ **do**
11:       Set $count = 0$
12:       **while** $count \leq N_t$ **do**
13:          Generate synthetic data sample $(x_j, y_j)$ using pretrained text-to-image diffusion model $\theta$
14:          Compute $\ell_g^t(x_j, y_j)$, $\ell_i^t(x_j, y_j)$, $s_{i,j}$, $\hat{p}_g^t(x_j)$
15:          **if** $s_{i,j} > 0$ **and** $\hat{p}_g^t(x_j) > \tau$ **then**
16:             Assign $(x_j, y_j)$ to client $i$ for supplementation: $(x_j, y_j) \in \tilde{\mathcal{D}}_i$
17:             $count = count + 1$
18:          **end if**
19:       **end while**
20:       Distribute $\tilde{\mathcal{D}}_i$ to client $i$
21:    **end for**Server train lora for clients:
22:    `DPPO Training`$(w_i^t, w_g^t, \tilde{\mathcal{D}})$
23:    `Client executes:`
24:    Update local model $w_i^t = w_g^t$
25:    Merge Dataset: $\mathcal{D}_i = \mathcal{D}_i \cup \tilde{\mathcal{D}}_i$
26:    $t = t + 1$
27: **end while**
28: **return** global model $w_g^T$

## A.2 Proof of Prediction Discrepancy as an Indicator of Label Scarcity

**Assumptions.** All clients share the class-conditional densities $q(x \mid y)$, while only the class priors differ: $\mathcal{P}_i(x, y) = d_{i,y} q(x \mid y)$ and $\mathcal{P}_g(x, y) = d_{g,y} q(x \mid y)$. Let $m_i(x) = \sum_{k=1}^C d_{i,k} q(x \mid k)$ and $m_g(x) = \sum_{k=1}^C d_{g,k} q(x \mid k)$. We use cross-entropy loss $\ell(w; x, y) = -\log p_w(y \mid x)$.

**Step 1: Bayes posteriors for every $x$.** By Bayes' theorem, for every $x$ and $y$,

$$p_i(y \mid x) = \frac{d_{i,y} q(x \mid y)}{\sum_{c=1}^C d_{i,c} q(x \mid c)} = \frac{d_{i,y} q(x \mid y)}{m_i(x)}, \qquad p_g(y \mid x) = \frac{d_{g,y} q(x \mid y)}{\sum_{c=1}^C d_{g,c} q(x \mid c)} = \frac{d_{g,y} q(x \mid y)}{m_g(x)}. \tag{12}$$

**Step 2: Strictly proper scoring rules imply Bayes-consistency.** Cross-entropy (log loss) is a strictly proper scoring rule (Gneiting & Raftery, 2007; Nguyen et al., 2010). Hence, under sufficient model capacity and optimization convergence, minimizing the cross-entropy risk on $\mathcal{P}_i$ yields the Bayes-optimal predictor:

$$p_{w_i}(\cdot \mid x) = \arg \min_{r(\cdot|x) \in \Delta_C} \mathbb{E}_{(x,y) \sim \mathcal{P}_i} \big[ -\log r(y \mid x) \big] = p_i(\cdot \mid x) \quad \text{a.s.} \tag{13}$$

An analogous statement holds for the global model, $p_{w_g}(\cdot \mid x) = p_g(\cdot \mid x)$.

**Step 3: Expected discrepancy (and strict positivity under overlap).** Fix a class $c$ and write $q_c(\cdot) \equiv q(\cdot \mid c)$. Using the Bayes posteriors from Step 1,

$$\ell(w_i; x, c) - \ell(w_g; x, c) = -\log p_{w_i}(c \mid x) + \log p_{w_g}(c \mid x) = \log \frac{p_g(c \mid x)}{p_i(c \mid x)}$$

$$= \log \frac{d_{g,c}}{d_{i,c}} + \log \frac{m_i(x)}{m_g(x)}. \tag{14}$$

Taking expectation over $x \sim q_c$ and using $\mathrm{KL}(q_c \| m) = \mathbb{E}_{q_c}[\log q_c(x) - \log m(x)]$, we obtain

$$\mathbb{E}_{x \sim q(\cdot|c)}[\ell(w_i; x, c) - \ell(w_g; x, c)] = \log \frac{d_{g,c}}{d_{i,c}} - \mathrm{KL}(q_c \| m_g) + \mathrm{KL}(q_c \| m_i). \tag{15}$$

Now write $m_\alpha(x) = \alpha \, q_c(x) + (1 - \alpha) \, r(x)$ with some density $r$ supported on $\{q(\cdot \mid k)\}_{k \neq c}$. Define $F(\alpha) = \mathrm{KL}(q_c \| m_\alpha)$. A direct calculation gives

$$F'(\alpha) \;=\; -\int q_c(x) \, \frac{q_c(x) - r(x)}{\alpha q_c(x) + (1 - \alpha) r(x)} \, dx \;\leq\; 0, \tag{16}$$

with strict inequality whenever $q_c$ and $r$ overlap on a set of positive measure. Hence $F(\alpha)$ is nonincreasing in $\alpha$. If $d_{g,c} > d_{i,c}$ then $\mathrm{KL}(q_c \| m_g) \leq \mathrm{KL}(q_c \| m_i)$, and

$$\mathbb{E}_{x \sim q(\cdot|c)}[\ell(w_i; x, c) - \ell(w_g; x, c)] \;\geq\; 0. \tag{17}$$

Moreover, if the class-conditionals are not perfectly separable (i.e., $q(\cdot \mid c)$ overlaps with some $q(\cdot \mid k)$, $k \neq c$), the inequality is strict:

$$\mathbb{E}_{x \sim q(\cdot|c)}[\ell(w_i; x, c) - \ell(w_g; x, c)] \;>\; 0. \tag{18}$$

**Conclusion.** Under label scarcity in client $i$ ($d_{i,c} < d_{g,c}$), the expected cross-entropy loss of the client model on class $c$ exceeds that of the global model when class-conditionals exhibit nontrivial overlap; in the degenerate perfectly separable case, the expectation equals 0.

### A.3 PROOF OF GRADIENT UPPER BOUND FOR CONFIDENCE-AWARE FILTER

**Lemma 1** (Per-sample loss bound). *Under cross-entropy loss with softmax probabilities $p_g^t(\cdot \mid x)$ and the confidence-aware filter $\hat{p}_g^t(x)_y \geq \tau$, we have*

$$\ell_g^t(x, y) = -\log \hat{p}_g^t(x)_y \;\leq\; -\log \tau.$$

*Proof.* Immediate from $\hat{p}_g^t(x)_y \geq \tau$ and the definition of cross-entropy. □

**Lemma 2** (Logit-gradient norm bound). *Let $z = w_g^t(x) \in \mathbb{R}^{\mathcal{C}}$ be the logits and $p = \mathrm{softmax}(z)$. For softmax cross-entropy $\ell(z, y) = -\log p_y$,*

$$\nabla_z \ell = p - e_y, \quad \text{hence} \quad \|\nabla_z \ell\|_2^2 = (1 - p_y)^2 + \sum_{k \neq y} p_k^2 \;\leq\; 2(1 - p_y). \tag{19}$$

*Under the filter $p_y \geq \tau$, it follows that*

$$\|\nabla_z \ell\|_2^2 \;\leq\; 2(1 - \tau), \qquad \|\nabla_z \ell\|_2 \;\leq\; \sqrt{2(1 - \tau)}. \tag{20}$$

*Proof.* Since $\sum_{k \neq y} p_k = 1 - p_y$ and $0 \leq p_k \leq 1$, we have $\sum_{k \neq y} p_k^2 \leq \sum_{k \neq y} p_k = 1 - p_y$. Therefore $\|\nabla_z \ell\|_2^2 \leq (1 - p_y)^2 + (1 - p_y) \leq 2(1 - p_y)$ because $(1 - p_y)^2 \leq (1 - p_y)$ for $p_y \in [0, 1]$. Applying $p_y \geq \tau$ yields the stated bounds. □

**Lemma 3** (Parameter-gradient bound). *Let $\theta$ denote the parameters of $w_g^t$ and suppose the logit Jacobian is bounded as $\|J_\theta z(x)\|_{\mathrm{op}} \leq L$ for all $x$ in the filtered set. Then, for filtered samples ($p_y \geq \tau$),*

$$\|\nabla_\theta \ell\|_2 = \|J_\theta z(x)^\top \nabla_z \ell\|_2 \;\leq\; L \|\nabla_z \ell\|_2 \;\leq\; L\sqrt{2(1 - \tau)}. \tag{21}$$

*Consequently,*

$$\mathbb{E}[\|\nabla_\theta \ell\|_2^2 \mid \hat{p}_g^t(x)_y \geq \tau] \;\leq\; 2L^2(1 - \tau). \tag{22}$$

*Proof.* Chain rule and the previous lemma. □

**Corollary 1** (Variance control). *Let $g = \nabla_\theta \ell$ for a filtered sample. Then $\mathrm{Var}(g) \leq \mathbb{E}\|g\|_2^2 \leq 2L^2(1 - \tau)$. In particular, increasing $\tau$ monotonically tightens the variance bound.*

## A.4 DETAILS OF DIFFERENT SCHEDULER STRATEGIES

In this subsection, we provide the details of different scheduler strategies used for controlling the amount of synthetic data supplied to clients across communication rounds. Let $T$ denote the total number of communication rounds, $N_t$ denote the number of supplemental samples assigned to each client at round $t$ and $N_{total}$ is the total number of supplemental samples assigned to each client during training.

**Uniform scheduler.** Each client receives the same number of supplemental samples in every round:

$$N_t = \frac{N_{total}}{T}, \quad \forall t = 1, \ldots, T, \tag{23}$$

**Linear decay scheduler.** The number of supplemental samples decreases linearly with the round index:

$$N_t = N_{total} \cdot \frac{T - t + 1}{\sum_{s=1}^{T} s}. \tag{24}$$

**Exponential scheduler.** The number of supplemental samples follows a normalized power-law decay:

$$N_t = N_{total} \cdot \frac{1}{\sum_{s=1}^{T} s^{-\beta}} \cdot t^{-\beta}, \tag{25}$$

where $\beta > 0$ controls the decay rate, and the normalization term ensures that $\sum_{t=1}^{T} N_t = D_{total}$.

We conduct comparison experiments on the CIFAR-10 dataset. The dataset is partitioned across 10 clients, with each client holding 500 local training samples. To evaluate the effectiveness of different supply scheduling strategies, we set the total number of supplemental samples to 500, which are distributed to clients within the first 40 communication rounds. The visualization of the per-round supplemental allocation is shown in Figure **??**, which clearly illustrates the distribution pattern of each scheduler.

## A.5 DISCREPANCY PREFERENCE OPTIMIZATION TRAINING DETAILS

We train the LoRA model for each client following the procedure described in Section 3.3. Only the LoRA parameters are updated, ensuring efficient personalization while keeping the diffusion backbone shared across all clients.

In standard Direct Preference Optimization (DPO), it is common practice to first perform Supervised Fine-Tuning (SFT) on the pretrained model with a small amount of data, followed by preference optimization. However, both computational overhead and limited data availability constrain the effectiveness of SFT in our setting. To ensure that Diffusion fine-tuning begins with an initial discrepancy from the reference model, we introduce a non-zero initialization of the LoRA parameters (e.g., Gaussian initialization), so that the fine-tuned model and the reference model produce distinguishable outputs from the very beginning.

In our experiments, we adopt the Low-Rank Adaptation (LoRA) framework for efficient fine-tuning. Specifically, we configure LoRA with a rank of 4, and a scaling factor of 4. The adaptation is applied to the attention projection layers, including to_q, to_k, to_v, and to_out.0.

For computational efficiency and in line with our design that data supplementation is most critical in the early stages of training, we restrict DPO finetuning to only the first $T_0$ rounds. Empirically, we find that applying DPO training for merely 3-5 rounds is sufficient to yield improvements, without incurring significant additional overhead.

## A.6 IMAGE GENERATION DETAILS

Specifically, we use the predefined prompt set: "a photo of a {class}", "a blurry photo of a {class}", "a black and white photo of a {class}", and "a photo of a small {class}" to generate more diverse images.

## A.7 DIFFERENT GENERATIVE MODEL

Table 9: Accuracy comparison of different generative model.

| Generative Model | $\alpha = 0.1$ | $\alpha = 0.5$ | $\alpha = 1.0$ |
|---|---|---|---|
| Stable Diffusion v1-4 | 36.32 | **34.46** | 31.93 |
| Flux-Dev | **37.98** | 34.40 | **32.80** |

We adopt the Stable Diffusion v1-4 model to synthesize images, using the default hyperparameters provided in the official implementation. The generation process is guided by a DDIM scheduler with 50 inference steps, and a guidance scale of 7.5. For each prompt, we first generate image at the default resolution of $512 \times 512$ and then resize them to match the target datasets: $32 \times 32$ for CIFAR-10 and CIFAR-100, and $64 \times 64$ for Tiny-ImageNet. The sampling temperature is kept at its default value. Through preliminary testing, we found that directly generating from $32 \times 32$ or $64 \times 64$ noise leads to poor image quality; hence, we adopt this two-step procedure to ensure both fidelity and compatibility with datasets.

## A.8 USE OF LLM

Large Language Models (LLMs), specifically OpenAI's ChatGPT (GPT-4/5), were used to assist with translation and with polishing the grammar, wording, and fluency of the manuscript. All scientific ideas, experimental designs, analyses, and conclusions are solely the authors' original work. The LLM was not used to generate research content, results, or references. The authors reviewed and verified all text produced with LLM assistance to ensure accuracy and integrity.

