# OpenReview forum: "We Generate What You Need: Efficient Data Supplement via Model Prediction Discrepancy for Heterogeneous Federated Learning"
_ICLR.cc/2026/Conference — ICLR 2026 Conference Withdrawn Submission_

### Official Review · Reviewer_Fv9H · 2025-10-31

**Soundness:** 2
**Presentation:** 2
**Contribution:** 2
**Rating:** 4
**Confidence:** 3

**Summary:**

The paper addresses data heterogeneity in federated learning by proposing to generate synthetic data to align the client’s dataset with the global distribution. They argue that, in practice, global class distributions can be imbalanced, whereas prior work assumes a balanced distribution. Their method supplements clients with the required synthetic data by using the local model’s performance as an indicator of the data distribution it was trained on. They also use preference optimization to adapt the diffusion model to generate samples that are more likely to be useful for the client.

**Strengths:**

1)The problem that the paper attempts to mitigate is well motivated and clearly articulated.
2)The efficacy of the proposed method to align local and global distributions is well evaluated and presented in Section. 4.2 Case Study.
3)The usage of the model’s performance to reflect the data distribution it is trained on is intuitive and practical to use.

**Weaknesses:**

1) The Following Relevant works are not cited in the related work.
 a) Confusion-Resistant Federated Learning via Diffusion-Based Data Harmonization on Non-IID Data
 b) pFedGPA: Diffusion-based Generative Parameter Aggregation for Personalized Federated Learning
2)The reward function in Eq. 8 needs further explanation as it seems to contradict the motivation for the reward mentioned in lines 300-302.
3)The experiments and results section could have been more clearly presented, for example: the metric compared against in tables 2 and 3 is not mentioned, and the details of generating global class imbalanced data and the degree of global class imbalance are not clearly mentioned.
4)Prior work (FedETF) has shown that for some methods, increasing the local epoch count can decrease the global performance. The decision to use five epochs to train local models for baselines vs one epoch for the proposed method is not well clarified.

**Questions:**

1) Do all the baselines used also use 5 epochs to train their local models? If not, please clarify the reasoning for using 5 epochs to train local models in baselines vs 1 epoch in the proposed method.
2)What would be the compute overhead at the server compared with the baselines?

---

> ### Author Response · Authors · 2025-11-23
> **[1/2] Addressing weakness 1-3**
>
> # Weakness 1: About the missing relevant works in the related work section.
>
> Thank you for pointing out these references. We have carefully reviewed both works, including “Confusion-Resistant Federated Learning via Diffusion-Based Data Harmonization on Non IID Data”[1] and “pFedGPA: Diffusion-based Generative Parameter Aggregation for Personalized Federated Learning.”[2] We agree that they are relevant to our topic. In the revised version, we will add appropriate discussions and citations in the related work section to clarify how our method differs from and complements these approaches.
>
> [1] *Confusion-Resistant Federated Learning via Diffusion-Based Data Harmonization on Non IID Data*, NIPS 2024
>
> [2] *pFedGPA: Diffusion-based Generative Parameter Aggregation for Personalized Federated Learning*, AAAI 2025
>
> ---
> # Weakness 2: About Equation 8.
>
> Thank you for pointing out this issue. You are correct that the signs on $\hat{p}_i^t(x_j)$ and $\hat{p}_g^t(x_j)$ should be swapped. We apologize for this oversight, and we will correct the expression in the revised manuscript.
>
> We also confirm that the experiments were conducted using the correct formulation. The code included in the supplementary material reflects the correct reward computation, and therefore the reported results are not affected by this typo.
>
> ---
> # Weakness 3: More details about metric and experiment settings
>
> Thank you for the helpful suggestions. We apologize for the lack of clarity in the original presentation.
>
> For the evaluation metrics in Tables 2 and 3, we report the best accuracy achieved during training. For each dataset and each heterogeneity setting, the experiments were repeated three times with different random seeds, and we report the mean and standard deviation across these runs.
>
> Regarding the construction of globally imbalanced data, we adopt a standard Dirichlet-based partitioning strategy. For each client, the label proportion vector is sampled as:
> $\mathbf{q}_i \sim \text{Dirichlet}(\alpha \mathbf{1}_C).$
>
> The parameter $\alpha$ directly determines the imbalance severity:
>
> - When $\alpha$ is small, the variance  $\mathrm{Var}(q_i^{(c)}) = \frac{\alpha (C - 1)}{C^2 (C\alpha + 1)}$
>   becomes large, producing highly skewed and class-imbalanced local datasets.
>
> - When $\alpha$ is large, $\mathbf{q}_i$ concentrates near the uniform distribution, yielding nearly IID clients.
>
> Here, $C$ denotes the number of classes, $\mathbf{q}_i=(q_i(1),\ldots,q_i(C))$ is the label proportion vector of client $i$, where $q_i(c)$ represents the proportion of class $c$ on client $i$; $\alpha$ is the Dirichlet concentration parameter; $\mathbf{1}_C$ is a $C$-dimensional all-one vector; and $\mathrm{Var}(q_i(c))$ measures how much the class proportions fluctuate across clients.
>
> To simulate realistic evaluation conditions with imbalanced test data, we also adjust the test dataset so that its label distribution matches the global training imbalance. We will add these details to the revised manuscript.

---

> ### Author Response · Authors · 2025-11-23
> **[2/2] Addressing weakness 4, Question 1-2**
>
> # Weakness 4, Question 1: About the choice of local epochs for baselines and for our method.
>
> Thank you for pointing out this issue. The reason our method uses smaller local epochs is that it supplies additional training data to each client, which increases local training steps. So we lower the local epoch to ensure a fair comparison with baselines.
>
> To further demonstrate that our method does not benefit unfairly from using a smaller number of local epochs, we conducted two additional experiments:
>
> 1. **Adopt 5 local epochs for our method:** We observe that increasing the number of local epochs improves the performance of our method, as clients learn more knowledge and provide more accurate loss signals for data supplementation.
> 2. **Adopt 1 local epoch for all methods:**  When all methods are restricted to one epoch, FedDPD consistently outperforms the baselines. Some baselines exhibit a small performance gain in this setting, but the improvement is limited and does not affect the relative ranking.
>
> The results are summarized in the table below.
>
> ## Comparison of FedDPD with different local epochs
>
> | Method                       | 0.05           | 0.1            | 0.5            | 1.0            |
> | ---------------------------- | -------------- | -------------- | -------------- | -------------- |
> | FedDPD (local epoch = 1)     | 64.64±2.23     | 59.96±3.67     | 57.09±3.14     | 58.95±1.37     |
> | **FedDPD (local epoch = 5)** | **67.31±2.35** | **62.16±3.47** | **58.36±3.54** | **59.65±1.00** |
>
> ## Performance of all methods with local epoch = 1
>
> | Local Epoch = 1 | 0.05           | 0.1            | 0.5            | 1.0            |
> | --------------- | -------------- | -------------- | -------------- | -------------- |
> | FedAvg          | 50.54±1.51     | 50.30±4.48     | 51.90±2.27     | 55.84±1.16     |
> | FedProto        | 50.08±0.43     | 52.34±4.24     | 52.68±1.86     | 56.95±1.32     |
> | FedETF          | 45.13±1.77     | 46.13±3.68     | 49.46±3.44     | 51.15±2.31     |
> | FedFA           | 46.29±0.74     | 44.59±7.31     | 50.64±3.43     | 52.62±0.83     |
> | GenFedSD        | 64.18±1.28     | 59.28±4.18     | 55.11±2.95     | 57.15±1.65     |
> | GenFL           | 51.78±1.36     | 53.12±4.97     | 53.01±2.93     | 57.03±2.14     |
> | **FedDPD**      | **64.64±2.23** | **59.96±3.67** | **57.09±3.14** | **58.95±1.37** |
>
> ---
> # Question 2: Computation overhead at the server compared with the baselines
>
> Thank you for this question. To quantify the server-side computation cost, we compared the total server computation time across different methods. Using FedAvg as the baseline, the measured server-side time is shown in the table below.
>
> | FedAvg | FedFA  | Gen_SD | GenFL   | FedDPD |
> | ------ | ------ | ------ | ------- | ------ |
> | 10315s | +2.38% | +3.84% | +14.22% | +7.83% |
>
> It is true that our method introduces additional server computation compared with the baselines. However, the extra cost remains small and increases the server computation time by only **7.83%** relative to the baseline. At the same time, our method provides a clear and consistent improvement in model performance, and we believe that this gain justifies the additional computation on the server.
>
> For more details, please refer to our response to Reviewer kKa6 (Weakness 2), where we thoroughly analyze the server-side overhead.

---

> > ### Comment · Reviewer_Fv9H · 2025-11-25
> > **revising the score**
> >
> > The authors answered the questions raised earlier. Revised the rating score.

---

> > > ### Author Response · Authors · 2025-12-03
> > >
> > > We are glad to hear that most of your concerns have been addressed, and we truly appreciate your positive feedback and helpful suggestions throughout the review process.

---

### Official Review · Reviewer_kKa6 · 2025-11-01

**Soundness:** 2
**Presentation:** 3
**Contribution:** 3
**Rating:** 4
**Confidence:** 3

**Summary:**

This paper seeks to mitigate the challenge of data heterogeneity in Federated Learning. The authors argue that earlier approaches try to balance each client's data distribution, which might not be the most suitable approach, as the global data distribution is most likely not balanced. To address this, the authors propose a new method (dubbed FedDPD) "that dynamically adjusts the data supplementation process according to the true global distribution, rather than assuming it to be balanced". Their aim is to provide each client model with data that fills the gap between the local model and the global (unknown) data distribution. Experiments on CIFAR-10, CIFAR-100, and Tiny ImageNet show that FedDPD outperforms state-of-the-art baselines, including other generative methods. It also demonstrates alignment with the true global distribution (Table 4).

**Strengths:**

The paper's primary strength is its challenge to the common "balance-oriented" augmentation assumption. It correctly identifies that real-world global distributions are often inherently imbalanced, posing a more realistic and important research question for the FL community.

The FedDPD framework is comprised of three main components:

**Prediction Discrepancy Selection (PDS):** This is the core idea. The server selects a synthetic sample $(x,y)$ for a client $i$ if the client's local model performs poorly compared to the global model, i.e. $\mathcal{l}_i​(x)>\mathcal{l}_g​(x)$. This discrepancy indicates the client's local data is deficient in samples of this type compared to the global data distribution. This is motivated by Theorem 1. In other words, the performance of the client compared to the server model is a proxy for data deficiency in the client models. Additionally, they implement the criterion that the softmax probabilities of the label have to be better than random, $\tau = \frac{1}{C}$ (they call this "confidence aware filtering")

**Exponential Decay Scheduler (EXS):** The paper finds that the _amount_ of data supplied per round matters. After testing several strategies (ablation results in Table 1), it adopts an exponential decay scheduler, which supplies more data in the early rounds and less later on, achieving the best stability and performance.

**Discrepancy Preference Optimization (DPPO):** To further tailor generation, the framework fine-tunes a client-specific LoRA module for the diffusion model. It uses a DPO-based reward function that prefers samples with high local loss, low global loss, and disagreement between the local and global model predictions.

**Elegant and Theoretically-Grounded Mechanism:**
The core mechanism of using prediction discrepancy ($l_i > l_g$) as a proxy for distributional gaps is intuitive, creative, and well-motivated by the theoretical argument in Theorem 1. This approach cleverly avoids needing to know the true global distribution.

**Strong Presentation and Clarity:**
The paper is very well-written. The problem and solution are explained clearly, and the figures (especially Figures 1, 2, and 3) are high-quality and effective at illustrating the core concepts (although Figures 2 and 3 are not referenced in the text itself).

**Weaknesses:**

**Reproducibility:**
It seems that code implementation is missing, which makes reproducing the results challenging.

**Misleading and Incomplete Efficiency Analysis**
The paper only comments on the overhead on the clients, but does not highlight the added overhead on the server. The only mention is that it takes around 2 seconds on an A100 GPU to generate an image, "demonstrating that our framework is computationally feasible for real-world applications".

1. Server-side computation:
This server-side cost scales linearly with the number of clients (N), a critical metric in federated learning. In a system with a large number of clients, this method might not be realistic, even if the generation only takes 2 seconds on an A100 GPU
The server must:
    * **Run a diffusion model** to generate a large pool of synthetic images.

    *  **Calculate discrepancy scores** for generated samples. This requires performing a forward pass using _both_ the global model and _each client's_ local model. This core step scales at least linearly with the number of clients ($N$), becoming computationally expensive as N grows
    * **Train N separate LoRA modules** using Discrepancy Preference Optimization (DPO), one for each client. This finetuning cost also scales linearly with N
    * This combined workload, which requires substantial, specialized GPU resources, is not benchmarked in terms of wall-clock time or cost. The $O(N)$ scaling of both inference and finetuning suggests the framework's server-side demands would become a bottleneck in real-world FL scenarios with many clients. The authors should include information about this added cost in their manuscript.

1. Communication overhead:
Line 113-114: "no additional computational or communication overhead on the client side". That statement could be misleading.
In addition to uploading models, as I understand it, clients must now _download_ the personalized supplemental datasets in each round. This download cost could become a practical issue, yet it is never measured or discussed, making the claim of "no additional computational overhead for clients" incomplete.

Equation 6: Additionally, if understood correctly, inference must be completed at both the client level and server level during training to obtain the losses and softmax probabilities used. This would also add communication overhead, along with added computational costs at the client.
If so, I suppose this would merit a more rigorous analysis of the claims.

1. **Domain shift of diffusion model**
The paper fails to address the generality of its generator-based approach. The method's success on common benchmarks (CIFAR, ImageNet) may be an artifact of data overlap, where the generator (Stable Diffusion) has already been trained on massive datasets containing similar images. This raises a critical question: Would the method still work for niche, domain-specific datasets (e.g., medical, satellite, or industrial images) that are _not_ well-represented in the generator's pre-training data? The paper provides no evidence for this, limiting the perceived generality of its contribution.

1. **Top/Bottom half cutoff for preferred/rejected (L306-308)**:
The DPPO component's reliance on a hard 50/50 cutoff to distinguish "preferred" from "rejected" samples is a methodological weakness. This arbitrary heuristic risks creating a noisy training signal by forcing the model to distinguish between samples with nearly identical reward scores that fall on opposite sides of this median split. This binary labeling discards the nuance of the continuous reward scores and may lead to mislabeled preferences, as the true proportion of "high-quality" samples generated in any given round is not guaranteed to be 50%.

1. Textual improvements
    *  Eq. 7, $N_{total}$ and Line 263 $N_{totl}$ should be the same
    *  Figures 2 and 3 do not seem to be referenced in the text.
    *  The text in Figures 4a and 4b is almost impossible to read in a printed version, because it is very small.
    * The conclusion is very short and is more of a summary.

**Questions:**

1. Line 46: newest reference is from 2022; newer work exists that should be cited.

1. Sec 2.2: A similar method to yours seems to be "Synthetic data shuffling accelerates the convergence of federated learning under data heterogeneity", Li et al, TMLR 2024, is that correct? If yes, should it be included as a baseline?

1. Eq. 3: $q(\cdot | c)$ does not seem to be defined in the main paper.

1. Line 199 (and further), $\hat{D}$ hat notation usually means estimator, so perhaps use a tilde instead?

1. Eq. 4: does not follow the definition of $\mathcal{l}$ in line 188. I think you meant to write: $\mathcal{l}(w_i; x_j,y_j) - \mathcal{l}(w_g; x_j,y_j)$ ?

1. For Tables 1-4, how many trials were run, and are the improvements significant after statistical tests? In Table 2, there is a clear overlap between your method's performance and baselines, which makes it unclear whether you actually have a significant improvement compared to the baseline.

1. Hyperparameter stability: How sensitive is the method to the choice of exponential scheduler coefficient $\beta$, confidence threshold $\tau$, and LoRA/DPO-specific settings? Can the authors share tuning curves or robustness analyses?

1. Is there any internal or external validation (human rating, metrics, FID, IS, etc.) of the diffusion-generated samples per class and per client?

1. Eq. 8: Should the sign on $\hat{p}_i^t(x_j)$  and $\hat{p}_g^t(x_j)$ not be swapped? You state that the goal is to encourage samples that are difficult for the client but easy for the global model. As I understand it, this would imply that $\hat{p}_g^t(x_j)> \hat{p}_i^t(x_j)$. But right now, the term $\hat{p}_i^t(x_j) - \hat{p}_g^t(x_j)$ would result in a negative reward if that is the case. In other words, by maximizing $\hat{p}_i^t(x_j) - \hat{p}_g^t(x_j)$ it seems that the framework would be selecting samples where $\hat{p}_i^t(x_j)> \hat{p}_g^t(x_j)$.

1. Line 296 + Eq. 9: Could logits be used instead of the class label?

1. Line 308: $\mathcal{S}_i^t = (x^+, x^-,y)$ is defined as a single pair, but in equation 11, you sum over pairs, and the text correctly states that it is a set of pairs. I think e.g in Eq. 11, $\mathcal{S}'$ should be defined as the set of all pairs?

1. Line 311: $\pi_{x|y}$ is the likelihood of the reference model, not "the reference model"?

---

> ### Author Response · Authors · 2025-11-23
> **[1/5] Addressing weakness 1-2**
>
> # Weakness 1: The code implementation is missing.
>
> Thanks for your review. We would like to clarify that our source code has been **provided in the supplementary material.** We are sorry that we do not specify it in our main paper. We commit to releasing the source code if the paper can be accepted.
>
> ---
> # Weakness 2: Concerns about server-side computation overhead
>
> We sincerely thank the reviewer for the insightful comments. We acknowledge that, in the current version of FedDPD, the server-side inference and the LoRA finetuning components scale as $O(N)$ with the number of clients. Following your suggestion, we performed a detailed profiling analysis of the server-side workload. We found that, under most practical FL settings, the additional overhead remains within a reasonable range. For more extreme FL settings where $N$ becomes very large, we additionally provide a solution: the server trains a **single shared LoRA adapter** instead of $N$ LoRA adapters. This design reduces the LoRA training cost from $O(N)$ to $O(1)$, a complexity reduction in server-side computation. Experiments show that this variant incurs a slight accuracy drop, but still outperforms all baselines.
>
> Our detailed analysis and additional experiments are provided below:
>
> ## Measured server-side overhead in our implementation
>
> We first provide runtime measurements of our original implementation, measuring three modules on the server. We conduct the following experiments on CIFAR-10 with 10 clients, the same settings as in the paper. The server is equipped with eight A100 80GB GPUs, and each client uses a single A100 GPU. This setup simulates a cross-silo scenario, where the number of clients is small and each client typically has strong computational resources.
>
> 1. **Image Generation:**  Due to the exponential decay scheduler, early rounds require more samples. We batch the prompts and feed them into the SD model. The runtimes of the first five rounds are:  **37.19 s, 18.75 s, 12.70 s, 9.96 s, 8.13 s**. Note that the average per-round client training time is **8.63 s**, so only the earliest rounds introduce little additional time cost. The system overall training time is 11122s. Overall, image generation only accounts for **≈0.7%** of the total training time.
>
> 2. **Local and global model Inference:**  Local and global forward passes are fast, averaging only **0.0075 seconds** across all epochs. This cost is nearly negligible when compared with client-side training.
>
> 3. **DPO Training:**  This part takes **176.72 s** per epoch. However, DPO is used only in the early epochs.
>    The system overall training time is 11122s, so its overall contribution is about **4.76%** of total training time.
>
>
> ## Scaling behavior with respect to the number of clients $N$
>
> Following the reviewer's concerns about linear growth with the number of clients, we increase $N$ to **30** and **50** while keeping all configurations unchanged. We find that for moderate values of $N$, the additional overhead introduced by our method remains at a reasonable level.
>
> 1. **Image Generation:** Synthetic images are shared by all clients. This stage remains constant with respect to $N$.
>
> 2. **Local/Global model Inference:** Although the computation increases with $N$, the inference cost is very small according to the analysis above. Therefore, the additional cost remains negligible compared with client-side local training.
>
> 3. **DPO Training:** The measured DPO execution times are listed below.
>
> |                                            | 10 clients | 30 clients | 50 clients |
> | ------------------------------------------ | ---------- | ---------- | ---------- |
> | **Time per epoch (s)**                     | 176.72     | 413.92     | 821.28     |
> | **ratio (dpo time / total training time)** | 4.76%      | 11.16%     | 22.15%     |
>
> Notably, the experiments reported above assume comparable computational capacity for both the server and the clients. In real-world deployments with a large number of clients, client devices typically possess weaker compute resources than the server. As a result, the actual ratio of additional server-side cost is expected to be even smaller than what is reported above.
>
> ## Computationally Efficient FedDPD
>
> For extreme FL settings where $N$ becomes very large, we introduced a solution: DPO trains a **single shared LoRA module** for all clients, using the (generated sample, reward) pairs collected from each client. The data are sorted by reward values, and we retain the same number of samples used in the original per-client DPO training in order to match the computation budget.
>
> We then evaluated the performance of this solution on Cifar-10, $N=100$, and observed a slight accuracy drop. Nevertheless, it still outperforms all baseline methods.

---

> ### Author Response · Authors · 2025-11-23
> **[2/5] Add weakness 2 share LoRA results, addressing weakness 3-5**
>
> The results of Computational Efficient FedDPD:
> | Method              | 0.05      | 0.1       | 0.5       | 1.0       |
> | ------------------- | --------- | --------- | --------- | --------- |
> | FedAvg              | 64.47     | 65.40     | 58.64     | 64.30     |
> | FedFA               | 58.22     | 62.40     | 59.24     | 62.98     |
> | GenFL               | 66.40     | 67.27     | 59.78     | 64.90     |
> | **FedDPD**          | **77.58** | **73.82** | **64.84** | **70.14** |
> | FedDPD (share LoRA) | 76.38     | 71.65     | 63.51     | 69.24     |
>
> ---
> # Weakness 3: Concerns about Communication overhead
>
> Thank you for pointing this out. You are correct that delivering supplemental datasets from the server to each client introduces additional **downlink** communication. Our original statement was referring specifically to the **uplink**, which is typically the bottleneck in FL systems.
>
> In standard FL deployments, the downlink bandwidth is substantially larger than the uplink bandwidth, and thus downlink transmissions rarely limit overall training throughput. This is why prior FL work (e.g., **FedAvg, FedProx, SCAFFOLD, MOON**) also focuses primarily on uplink efficiency.
>
> We will revise the paper to explicitly state that FedDPD introduces **no additional uplink communication overhead**, while the added downlink cost is lightweight and does not impact system bottlenecks in practice.
>
> ---
> # Weakness 4: About Equation 6 and the associated communication cost
>
> Thank you for your comments. We would like to clarify that all computations in Equation (6), including the loss evaluation and the softmax confidence of both the global model and each client model, are performed **entirely on the server side**. The server uses the models uploaded by clients to evaluate synthetic samples that are also generated on the server.
>
> Clients do not receive inference requests and do not participate in any computation related to discrepancy scoring. The only information clients obtain is the selected supplemental data.
>
> For this reason, **Equation (6) does not introduce any additional client-side computation or communication** beyond the **standard model upload** step that already exists in FedAvg-style FL. We will clarify this point in the updated manuscript.
>
> ---
> # Weakness 5: About the domain shift of the diffusion model
>
> Thank you for raising this important concern. To evaluate whether FedDPD remains effective on domains that are not well represented in the pretraining data of the Stable Diffusion model, we conducted two additional experiments on the **satellite domain** (EuroSAT) and the **medical domain** (ChestMNIST), respectively. These datasets represent domains that are very different from the natural images seen by the generator during pretraining.
>
> We compared FedDPD with state-of-the-art FL methods and the results are shown in the table below. We analyze the results from three perspectives:
>
> 1. **High class-imbalance** scenarios (lower alpha): Our method continues to outperform all baselines even under strong global imbalance. This is because FedDPD effectively addresses the heterogeneity introduced by class imbalance. Therefore, despite the inherent domain gap between diffusion-generated and real data, our framework maintains the best performance.
> 2. **Low class-imbalance** scenarios (higher alpha), compared with **generative baselines**: Even in low-imbalance settings, our method still outperforms generation-based augmentation approaches such as GenFL and Gen-FedSD. All generative methods are affected by the domain gap, but our framework better mitigates the heterogeneity caused by class imbalance, allowing it to retain superior performance.
> 3. **Low class-imbalance** scenarios (higher alpha), compared with **non-generative baselines**: In settings with mild imbalance, our method is slightly weaker than several non-generative baselines. When class imbalance is low, the negative effect of the domain gap exceeds the gain brought by alleviating class imbalance, which leads to a modest performance drop.
>
> ## EuroSAT
>
> | Method/alpha | 0.05      | 0.1       | 0.5       | 1.0       |
> | ------------ | --------- | --------- | --------- | --------- |
> | FedAvg       | 51.74     | 61.92     | 82.09     | 85.94     |
> | FedFA        | 59.04     | 75.89     | 89.23     | 90.24     |
> | FedProto     | 58.99     | 77.19     | 90.98     | **90.50** |
> | GenFL        | 59.13     | 74.51     | 86.57     | 81.20     |
> | Gen_SD       | 63.29     | 78.40     | 87.14     | 85.33     |
> | **FedDPD**   | **67.30** | **82.46** | **91.10** | 87.62     |

---

> ### Author Response · Authors · 2025-11-23
> **[3/5] Add weakness 5 ChestMNIST results, addressing weakness 5-7, question 1**
>
> ## ChestMNIST
>
> | Method/alpha | 0.05      | 0.1       | 0.5       | 1.0       |
> | ------------ | --------- | --------- | --------- | --------- |
> | FedAvg       | 24.82     | 21.90     | 17.52     | 17.47     |
> | FedFA        | 24.33     | 22.29     | 18.19     | **18.15** |
> | FedProto     | 25.14     | 22.10     | 18.82     | 18.11     |
> | GenFL        | 25.06     | 21.26     | 18.62     | 17.35     |
> | Gen_SD       | 25.73     | 21.73     | 18.84     | 17.16     |
> | **FedDPD**   | **27.13** | **23.27** | **19.29** | 17.91     |
>
> In conclusion, domain shift scenario introduces a trade-off between **class imbalance** **heterogeneity** and **domain gap**. On the one hand, our method effectively alleviates class imbalance, which brings clear positive gains. On the other hand, the synthetic data generated by the diffusion model exhibits a domain gap relative to the original training data, which can introduce negative effects. Our experiments show that in most heterogeneous settings, the positive gains outweigh the negative impact, leading to performance improvements. We claim that, under domain-shift settings, the scope of our method targets scenarios with moderate or high level of heterogeneity. For practical application, our methods can easily extend to various domains by replacing the diffusion model to a domain-specific data generator.
>
> ---
> # Weakness 6: Top/Bottom half cutoff for preferred/rejected
>
> We thank the reviewer for the thoughtful comments. The concern highlights two important issues, and we provide clarification for both points.
>
> 1. **Tackling nearly identical reward scores:** Regarding the possibility that two samples may have nearly identical reward scores, we agree that a strict fifty–fifty split can create ambiguous preference pairs. In our implementation, we explicitly ignore sample pairs whose reward difference is smaller than a predefined margin. This avoids constructing preference pairs that carry little meaningful signal. This detail was omitted in the paper for brevity. We will add a clear explanation in the revised version.
>
> 2. **Learning relative relations instead of the samples themselves:** We agree that the true proportion of high-quality samples produced in a generation round is not necessarily fifty percent. However, DPO only relies on learning *relative* relations. Even if some low-quality samples are included in the positive set, the corresponding negative samples are **lower-quality** samples, which still provides the model with a correct direction to adjust its generative bias.
>
> To further address the reviewer’s concern, we conducted an additional experiment in which we replaced the median split with a general Top $p$ percent versus Bottom $p$ percent strategy, where $p \in \{0.2, 0.3, 0.4, 0.5\}$. The results are shown below.
>
> | Ratio 0.2 | Ratio 0.3 | Ratio 0.4 | Ratio 0.5 |
> | --------- | --------- | --------- | --------- |
> | 63.30     | 63.84     | 63.24     | **64.14** |
>
> These results indicate that the median split is an empirically strong and stable choice, and the performance difference across different ratios is small. We will add this robustness analysis in the revised manuscript.
>
> ---
> # Weakness 7: Textual improvements.
>
> Thank you for pointing out these issues. We appreciate the careful reading. We will thoroughly proofread the manuscript and correct all noted problems in the revised version. This includes ensuring consistent notation in Equation (7) and Line 263, adding explicit references to Figures 2 and 3 in the main text, enlarging the font in Figures 4a and 4b to ensure readability in print, and expanding the conclusion to provide a clearer and more complete discussion. We thank the reviewer for helping us improve the clarity and presentation quality of the paper.
>
> ---
> # Question 1: About outdated references in the early part of the related work section.
>
> Thank you for pointing this out. We agree that related work should reflect recent advances in federated learning. We will add several newly published works, including “Adaptive Self Distillation for Minimizing Client Drift in Heterogeneous Federated Learning” (TMLR 2024), “Federated Learning with Sample Level Client Drift Mitigation” (AAAI 2025), and “Debiasing Federated Learning with Correlated Client Participation” in (ICLR 2025) to our revised version. These methods also aim to address the model drift issue in FL, and we will add a clear discussion of their relevance to our work in our revised paper.

---

> ### Author Response · Authors · 2025-11-23
> **[4/5] Addressing question 2-6**
>
> # Question 2: A similar work published in TMLR 2024
>
> Thank you for highlighting this work. After carefully reviewing the paper, we find that its assumption and methodology differ substantially from ours.
>
> 1. **Difference in assumption**: Their approach supplements client data under the assumption of a balanced global distribution, while our method removes this assumption and can work whether global label distribution is balanced or not.
> 2. **Difference in methodology**: Their method trains its own generative model using local data and then generates synthetic samples for other clients. Instead, we identify the distribution gaps through model prediction discrepancy and supplement each client with targeted synthetic data produced on the server.
>
> Nevertheless, since this method also involves synthetic data generation, we agree that including it as a baseline is meaningful. During the rebuttal period, we reproduced this method and conducted comparative experiments on the Cifar-10 dataset. The results show that our method achieves higher performance. Due to time limitations in the rebuttal stage, we will complete evaluations on all datasets and include these results in the revised manuscript.
>
> | Method      | 0.05           | 0.1            | 0.5            | 1.0            |
> | ----------- | -------------- | -------------- | -------------- | -------------- |
> | FedAvg      | 51.58±2.46     | 51.36±3.82     | 50.60±2.89     | 52.98±0.99     |
> | Fedssyn [1] | 52.51±2.78     | 49.56±4.02     | 48.44±2.94     | 50.31±0.68     |
> | **FedDPD**  | **64.64±2.23** | **59.96±3.67** | **57.09±3.14** | **58.95±1.37** |
>
> [1] *Synthetic data shuffling accelerates the convergence of federated learning under data heterogeneity*, TMLR 2024
>
> ---
> # Question 3 & 4 & 5: Clarification regarding notation inconsistencies
>
> Thank you for the careful reading and for pointing out these issues.
>
> - **Clarification for Equation (3)**: We acknowledge that the notation was unclear.  The term $q(\cdot \mid c)$ refers to the class-conditional data distribution for label $c$.  We will correct and explicitly define this notation in the revised manuscript to avoid ambiguity.
> - **Clarification for the notation $\hat{D}$** : We agree with the reviewer that the hat symbol is typically used to denote an estimator.  The tilde symbol is more appropriate for representing a synthetic dataset. We will update the notation throughout the paper to reflect this convention.
> - **Clarification for Equation (4)** : You are correct that the current formulation does not follow the definition in line 188.  The correct expression should be: $\ell(w_i, x_j, y_j) - \ell(w_g, x_j, y_j).$
>
> We will correct this in the revised version.  We appreciate these suggestions, which help improve the clarity and precision of the presentation.
>
> ---
> # Question 6: About the number of trials and the statistical significance of the reported improvements
>
> Thanks for your review. For each dataset and each heterogeneity setting, we perform three independent runs under identical experimental configurations. As the reviewer correctly noted, CIFAR-10 shows some overlap in the reported mean and standard deviation values. To address this, we conducted the following statistical significance tests comparing FedDPD with the strongest baseline on CIFAR-10:
>
> - **The paired t-test:** computes the paired difference for each run, takes the mean difference, and tests it against zero using the t-statistic based on the sample mean and sample variance. It evaluates whether the mean improvement of FedDPD is significantly greater than zero.
> - **The Wilcoxon signed-rank test:** It ranks the absolute paired differences, restores their signs, and tests whether the sum of positive ranks is significantly larger than the sum of negative ranks. It assesses whether this improvement is consistently positive without assuming normality.
> - **The sign test:** It counts how many times FedDPD outperforms the baseline (wins) versus underperforms (losses), and evaluates whether the observed imbalance in signs could arise from a fair Bernoulli process with probability 0.5.
>
> For all three tests, they calculates a p-value. **p-value below 0.05** indicates that the observed improvement is unlikely to be due to random fluctuations. We find that the paired t-test, the Wilcoxon signed-rank test, and the sign test yield  **$p=1.2 \times 10^{-4}, p = 0.0024, p = 0.019$**.  **All values are well below the 0.05 threshold, indicating that the improvement is statistically significant.**
>
> These results confirm that, despite the visual overlap of mean and standard deviation ranges, FedDPD consistently outperforms the strongest baseline on CIFAR-10. We will include these statistical tests in the revised manuscript.

---

> ### Author Response · Authors · 2025-11-23
> **[5/5] Addressing question 7-12**
>
> # Question 7: About Hyperparameter stability
>
> Thank you for raising this question. To evaluate the sensitivity of our method to hyperparameters, we conducted additional experiments that varied the key parameters used in our method. Specifically, we sample:
>
> - exponential decay coefficient $\beta \in \{0.8, 0.85, 0.9, 0.95, 1.0, 1.05, 1.1\}$
> - global model confidence threshold $\tau \in \{0, 0.03, 0.05, 0.08, 0.1, 0.12, 0.15\}$
> - DPO disagree reward coefficient $\lambda \in \{0.2, 0.3, 0.4, 0.5, 0.6, 0.7, 0.8\}$
>
> The results are presented in the tables below.
>
> | $\beta = 0.8$ | $\beta = 0.85$ | $\beta = 0.9$ | $\beta = 0.95$ | $\beta = 1.0$ | $\beta = 1.05$ | $\beta = 1.1$ |
> | ------------- | -------------- | ------------- | -------------- | ------------- | -------------- | ------------- |
> | 62.34         | 63.84          | 63.06         | 64.08          | **64.14**     | 63.48          | 63.54         |
>
> | $\tau = 0$ | $\tau = 0.03$ | $\tau = 0.05$ | $\tau = 0.08$ | $\tau = 0.1$ | $\tau = 0.12$ | $\tau = 0.15$ |
> | ---------- | ------------- | ------------- | ------------- | ------------ | ------------- | ------------- |
> | 61.74      | 61.80         | 62.88         | 62.52         | **64.14**    | 62.19         | 60.67         |
>
> | $\lambda = 0.2$ | $\lambda = 0.3$ | $\lambda = 0.4$ | $\lambda = 0.5$ | $\lambda = 0.6$ | $\lambda = 0.7$ | $\lambda = 0.8$ |
> | --------------- | --------------- | --------------- | --------------- | --------------- | --------------- | --------------- |
> | 62.34           | 62.16           | 63.80           | **64.14**       | 63.25           | 60.96           | 62.22           |
>
> ---
> # Question 8: About external validation of the diffusion generated samples.
>
> Thank you for the question. In our framework, we do not introduce additional external validation such as human evaluation, FID, or IS scores for the synthesized samples. Our method generates data using a standard Stable Diffusion model, and the usefulness of each sample is determined entirely through two internal signals.
>
> The first signal is the confidence of the global model, and the second is the prediction discrepancy between the local and global models. These signals directly reflect whether a synthetic sample can help reduce the local–global distribution gap, which is precisely the objective of our method.
>
> ---
> # Question 9: About Equation 8.
>
> Thank you for pointing out this issue. You are correct that the signs on $\hat{p}_i^t(x_j)$ and $\hat{p}_g^t(x_j)$ should be swapped. We apologize for this oversight, and we will correct the expression in the revised manuscript.
>
> We also confirm that the experiments were conducted using the correct formulation. The code included in the supplementary material reflects the correct reward computation, and therefore the reported results are not affected by this typo.
>
> ---
> # Question 10: About using logits in Eq. 9
>
> Thank you for the insightful suggestion. We agree that the disagreement term in Eq. 9 could, in principle, be replaced with a soft metric about logits, such as the KL divergence between the local model’s and global model’s probability outputs, thereby creating a “soft reward” that reflects the degree of mismatch between the two models.
>
> However, this formulation introduces a potential issue: probability-based distances capture full-distribution differences, not only class-level disagreement. For example, consider two predictions where the local and global models assign the same top-1 class:
>
> - local logits:  [0.1, 0.5, 0.4]
> - global logits: [0.3, 0.7, 0.0]
>
> Although the top-1 prediction is identical, the KL divergence between these distributions is non-trivial, which may introduce noise into the reward computation. This is why our current formulation focuses solely on class-level disagreement.
>
> We appreciate the suggestion and will explore this variant in future work.
>
> ---
> # Question 11 & 12: About textual improvements
>
> Thank you for pointing this out. These issues are indeed due to our writing oversight:
>
> - $S_i^t$ should denote the set of positive–negative preference pairs.
> - $\pi_{\text{ref}}(x \mid y)$ refers to the likelihood under the reference model.
>
> We will correct both notations in the revised manuscript.

---

> > ### Comment · Reviewer_kKa6 · 2025-11-26
> >
> > Thank you for all your updates and hard work to accommodate the questions and weaknesses. These largely address most of my concerns.
> >
> > I have serious doubts about the table containing FedSyn, and in the response to reviewer croB, FedDifRC and DPSDA-FL. I quickly checked the results reported in the respective papers, and if I am not mistaken, they all perform much better than FedAvg. However, in your table, the methods are on par with FedAvg in almost all cases. FedDifRC (alpha=0.05) and Fedssyn (alpha={0.1, 0.5, 1.0}) even perform worse. How can this be?
> >
> > Did you not have time (which is understandable) to tune the methods properly, or are the settings in the respective papers and your paper significantly different, or something else?

---

> > > ### Author Response · Authors · 2025-12-03
> > >
> > > Thank you for the question. For **DPSDA-FL and Fedssyn**, the reason for the discrepancy between our results and those reported in their papers is that **our experimental setting differs from theirs.** Our experiments are conducted under *imbalanced* global data distributions, while these approaches are evaluated under balanced global distribution in their paper. Under a globally imbalanced dataset, DPSDA-FL and Fedssyn supply each client's data toward a balanced distribution actually increases the divergence between the local and true global distributions, which in turn degrades performance and in some cases even yields results worse than FedAvg.
> > >
> > > For **FedDifRC**, which evaluates globally imbalanced settings only at $\alpha = 0.2$, our results at $\alpha \in \{0.1,0.5,1.0\}$ are consistent with the findings reported in the original paper. Under the highly heterogeneous ($\alpha =0.05$) scenario, a possible explanation for the degradation is that FedDifRC is not robust to severe intra-client heterogeneity, and its performance drops noticeably as the heterogeneity increases. Similar trends can also be observed in the original paper.
> > >
> > > To further address your concern regarding the results of the SOTA baselines, we additionally conducted experiments under globally balanced data distributions, using CIFAR-10 and heterogeneity levels of 0.05, 0.1, 0.5, and 1.0 (three runs each), we observe that all three methods (DPSDA-FL, Fedssyn, FedDifRC) show clear performance improvements compared to the FedAvg.
> > >
> > > |          | $\alpha=0.05$  | $\alpha=0.1$   | $\alpha=0.5$   | $\alpha=1.0$   |
> > > | :------- | :------------- | :------------- | :------------- | :------------- |
> > > | FedAvg   | 37.11±3.62     | 40.88±2.85     | 47.41±0.78     | 49.10±0.60     |
> > > | Fedssyn  | 41.12±3.64     | 45.65±1.05     | 50.63±0.63     | 51.88±0.86     |
> > > | DPSDA-FL | 44.89±3.86     | 47.97±2.39     | 51.81±1.03     | 53.47±0.94     |
> > > | FedDifRC | 45.56±3.49     | 48.58±2.92     | 52.47±0.51     | 53.04±0.77     |
> > > | FedDPD   | **46.95±3.99** | **50.78±1.58** | **53.19±1.34** | **54.14±1.02** |
> > >
> > > Notably, our method does not rely on whether the global distribution is balanced or imbalanced, and consistently achieves the best performance across all evaluated scenarios.

---

### Official Review · Reviewer_croB · 2025-11-01

**Soundness:** 3
**Presentation:** 3
**Contribution:** 2
**Rating:** 6
**Confidence:** 3

**Summary:**

To address data heterogeneity in Federated Learning, diffusion models can generate synthetic data to align local and global distributions. Existing methods assume a balanced global distribution, but real-world data is often naturally imbalanced. The authors propose FedDPD, which uses performance discrepancies between local and global models to identify distribution gaps and generates tailored synthetic data via optimized diffusion models. Without adding client-side computation, FedDPD outperforms state-of-the-art methods by up to 3.82% across varied global distributions.

**Strengths:**

The paper is well-structured and easy-to-follow. The proposed approach outperformed the listing benchmarking algorithms.

**Weaknesses:**

More recent and state-of-the-art sample generation methods should be taken in to consideration and comparison. I noticed there are two generative FL algorithm in Table 2 and 3. Nevertheless, more recent approaches should be involved.

**Questions:**

Please refer to weaknesses for my main concerns.

---

> ### Author Response · Authors · 2025-11-23
>
> # Weakness 1: More state-of-the-art generation methods should be compared
>
> Thanks for your insightful comments. We agree with the reviewer's point that we need to add more generation methods for comparison. In response to your concern, we additionally include three recently published state-of-the-art methods \[1\]\[2\]\[3\] for comparison. The results on CIFAR-10 are shown below. We can see that our FedDPD can consistently outperform these methods, demonstrating the effectiveness of our method.
>
> | Method/Alpha     | 0.05           | 0.1            | 0.5            | 1.0            |
> | ---------------- | -------------- | -------------- | -------------- | -------------- |
> | **FedAvg**       | 51.58±2.46     | 51.36±3.82     | 50.60±2.89     | 52.98±0.99     |
> | **FedDifRC** [1] | 49.02±1.98     | 52.60±5.57     | 55.05±3.26     | 57.59±1.17     |
> | **DPSDA-FL** [2] | 53.17±0.80     | 53.29±4.26     | 52.40±3.18     | 55.94±0.81     |
> | **Fedssyn** [3]  | 52.51±2.78     | 49.56±4.02     | 48.44±2.94     | 50.31±0.68     |
> | **FedDPD**       | **64.64±2.23** | **59.96±3.67** | **57.09±3.14** | **58.95±1.37** |
>
> **References:**
>
> [1] *Unlocking the Potential of Text-to-Image Diffusion Models in Heterogeneous Federated Learning*, ICCV 2025
>
> [2] *Synthetic Data Aided Federated Learning Using Foundation Models*, arxiv 2025
>
> [3] *Synthetic data shuffling accelerates the convergence of federated learning under data heterogeneity*, TMLR 2024

---

### Official Review · Reviewer_zK2q · 2025-11-15

**Soundness:** 3
**Presentation:** 3
**Contribution:** 3
**Rating:** 6
**Confidence:** 4

**Summary:**

This paper is about methods to improve performance of a global model trained through federated learning on heterogenous data. Note that in the heterogenous FL setting, there is an inherent tradeoff between fitting a model that is good on the global distribution (the "global" setting) and maximizing performance on a per-client basis (the "personalized" setting). This paper focuses on the global setting, where the goal is to introduce regularization that prevents client models from overfitting their local data and diverging from one another. Early research on this problem focused on explicit regularization (e.g., FedProx, gradient clipping, etc) but more recently people have been looking at implicit regularization (e.g., data augmentation).

In this paper, the authors propose a method, FedDPD, that targets the problem of class imbalance for federated multi-class classification tasks. The core idea is that we can identify class deficiencies in a client by examining the performance difference between the global and client models (I think there are some problems with this claim in general, but at least it seems to be true in practice).

The algorithm is to (1) collect client model updates as usual in FL, (2) generate synthetic samples on the server, (3) measure the performance difference between the global model and each client model for each sample, and (4) update the client models on the server (before aggregation) using the synthetic examples where there was a large difference in loss between the client and global models. The algorithm additionally updates the generative model using DPO, to improve the likelihood that it will generate samples where the local and global models have high discrepancy.

**Strengths:**

- The problem is important and has lots of practical applications.
- It is a nice idea to do the synthetic data generation once, on the FL aggregation server, rather than to force clients to do this locally.
- The experiments show some performance improvements from FedDPD, both on its own and also when combined with other methods, and compare against a good number of baselines.
- The synthetic sample generation is effectively a rejection sampling algorithm, where we sample from a generative model and then check that (1) the global model has high confidence and (2) the global and local models disagree. DPO is a very reasonable way to improve the sample efficiency of this process, by learning a model of the non-overlapping portions of the distributions.

**Weaknesses:**

- FedDPD is somewhat narrowly focused on multi-class federated classification tasks for data modalities where it is possible to construct a generative model (and specialize it through low-rank adaptation). Many important problems fall into this category, so this is not a huge weakness. Even still, the framework does not immediately generalize to other types of data heterogenity, such as feature drift, or other types of objective, such as generative modeling or regression (and this should probably be clarified in the paper).
- The label distribution correction behavior does not seem to be guaranteed. Figure 1a suggests that FedDPD augments the class distribution of each client to match the global distribution, but in reality we never actually see the global / local class distributions. Instead, we just see the losses of the client and global models on the synthetic dataset. Theorem 1 says that this performance difference will be negative (in expectation) if the client is deficient in some class. However, I am not sure about the converse - a poor loss from the client on an example where the global model succeeds does not necessarily indicate that the client is deficient in that class. For example, consider a situation where the client and global distributions have equal class proportions but have feature drift, or a situation where either the client or the global model have not converged. In these situations, FedDPD might make the class imbalance worse.
- Further, Theorem 1 seems to have some important assumptions that are not stated clearly in the text or satisfied in practice. If my understanding of the proof in A2 is correct, we require that feature drift does not occur (clients share $q(x|y)$), the client / global models are fully converged (so that the empirical cross-entropy risks are minimized). These assumptions seem unrealistic in the heterogenous FL setting, especially the ones that require convergence of the model, since heterogenity harms the convergence rate / stability.

Of course, the experiments show that these problems do not occur in practice (Figure 4). But the paper should probably discuss these caveats and make it clear that the theoretical development is to provide intuition, not guarantees.

There are also some minor issues / typos in writing
- The FedDPD acronym is not defined anywhere in the main text.
- "We employs a text-to-image diffusion model on the server" on L198

**Questions:**

Did you conduct an ablation where the DPO finetuning was removed? I wonder whether DPO is effectively learning a model of the distribution difference between - especially since DPO is only run for the first few rounds of FL. If so, then the algorithm in this paper is actually very similar to the one in *"Federated Learning in Non-IID Settings Aided by Differentially Private Synthetic Data"* - just instead of finding clients with complementary data distributions to merge with, we construct a complementary data distribution on the server using DPO.

---

> ### Author Response · Authors · 2025-11-23
>
> # Weakness 1: About the scope and generality of FedDPD.
>
> Thank you for raising this point.
>
> ## The scope of FedDPD
>
> We agree that FedDPD is specifically designed for multi class federated **classification tasks**, instead of other tasks, such as generative modeling or regression. We will clearly state this scope in the revised version.
>
> ## Extending to feature shift non-IID
>
> Thank you for your insightful comments. Regarding feature shift non IID scenarios, the current version of FedDPD assumes that clients share the same visual domain and differ mainly in the label distribution. Although current design does not explicitly address domain level feature shifts, the overall framework can be extended in principle. One simple solution is to modify the diffusion prompts to incorporate domain related information (e.g., A photo of a cat with cartoon style.) so that the generator can produce samples from different styles or domains. Once such domain aware generation is available, the loss discrepancy mechanism could be used to identify which domains a client lacks. Since this extension requires additional methodological components and lies beyond the core scope of the present work, we plan to explore it as future work.  We will explicitly discuss the scope and these limitations in the revised manuscript.
>
> ---
> # Weakness 2 and Weakness 3: About the reliability of the loss discrepancy signal and the assumptions in Theorem 1.
>
> Thank you for raising this concern. We understand that Weakness #2 and Weakness #3 reflect the same underlying concern, namely whether the loss discrepancy used by FedDPD remains reliable when factors such as feature shift or incomplete model convergence are present. We therefore provide a unified response to address both comments.
>
> We agree that the loss discrepancy between the local and global models serves as a proxy for identifying class deficiency rather than a strict guarantee. Theorem 1 shows that if a client indeed lacks samples from a certain class, the discrepancy tends to be positive for that class. However, the converse does not necessarily hold. A large discrepancy may also arise from other factors, such as feature shift or incomplete model convergence. For this reason, the theoreticals result should be understood as providing intuition for why the proxy is effective in label imbalance scenarios, rather than as a guarantee that applies to all forms of heterogeneity.
>
> We will make these assumptions, limitations, and empirical observations explicit in the revised manuscript and clarify the scope of the theoretical analysis.
>
> ---
> # Question 1: About DPO finetuning
>
> Thank you for the insightful question. We have conducted an ablation study on the DPO module, as presented in Section 4.4 and Table 8. The results demonstrate that DPO finetuning is indeed effective, even when applied for only a few epochs.
>
> Regarding *Federated Learning in Non-IID Settings Aided by Differentially Private Synthetic Data*, the key difference from our work is that their method still assumes a balanced global label distribution, whereas our framework does not rely on this assumption. Thank you for the helpful suggestion. We will include a discussion of this related work in the revised version of the paper.

---

> > ### Comment · Reviewer_zK2q · 2025-11-26
> >
> > Thank you for your responses.
> >
> > 1. I don't have any more concerns about the theory, provided that the limitations of Theorem 1 are called out more clearly in the revision (as providing intuition / supporting evidence but not rigorous proof). The paper's algorithm relies on the fact that the converse holds true in practice, since we use the loss difference to identify and correct label discrepancy. This is not implied by the theory but is instead demonstrated by the experiments, which are in my opinion the main contribution of the work.
> >
> > 2. Thank you for directing me to the ablation study, which I had missed. This addresses my concern about whether DPPO is doing all of the heavy lifting.
> >
> > 3. However after taking a more detailed look at these add-on experiments, I am somewhat confused. Is the setup for Table 8 directly comparable to the setup for Table 2? If so, then why is the performance of the full FedDPD method (with PDS/EXS/DPPO) several points lower in Table 8 than in the corresponding entry in Table 2? As an example, at alpha = 0.1, we have 55.44 in Table 8 but 59.96 +/- 3.7 in Table 2. This discrepancy seems to also hold for some of the other experiments (such as those in Table 6, where FedFA overperforms the equivalent setting in Table 2). Ideally, it would be possible to compare these results in the context of Table 2. I understand that it is likely too computationally expensive to do the full grid of ablations for combining methods, i.e. DPD + (FedAvg | FedProx | FedProto | ... ), and for ablating heuristics, i.e. (PDS, PDS+EXS, DPPO, ALL). But it would be good to have an idea, for example, of where FedDPD would fall in Table 2 if we left out the DPPO component or where FedFA + FedDPD would rank. In some cases the intervals reported in Table 2 do not contain the values reported in Table 6, 8.
> >
> > Note that the statistical significance analysis in your response to Reviewer kKa6 addresses concerns about whether the reported differences between methods in Table 2 are reliable.
> >
> > 4. Finally, I still view feature shift as a problem that is not easily solved by this paper. I think the ideas in the response will work well for image domains, where we can easily characterize drift as a style / format difference, and where we do not expect the support of one class distribution to collide with the support of another due to drift. But there are other diffusion-friendly settings, such as text or audio, where one might be able to apply the FedDPD idea but where this is not the case. For example, a text topic classification problem may experience significant feature drift as the meanings and associations of words change over time and across the client population. Here, the problem is much more nuanced.

---

> ### Author Response · Authors · 2025-12-03
>
> Thank you for the careful observation. We confirm that the experimental setup in Table 8 is consistent with the main setup in Table 2. **The discrepancy is due to the number of seeds**: 55.44 in Table 8 comes from a **single-seed run** (seed = 0), whereas 59.96 ± 3.67 in Table 2 is **averaged over three seeds** (seed = 0, 1, 2). Here we provide concrete results under seed 0, 1, 2: [55.44, 60.01, 64.43], whose mean and standard deviation exactly match the value reported in Table 2 (59.96 ± 3.67). The same situation explains the FedFA in Table 6.
>
> We also integrate these results into Table 2 as you suggested. We conduct multi-seed experiments for FedDPD without DPO and for the FedFA + FedDPD combination. The integrated results are shown below.
>
> | **Cifar10**    | **$\alpha = 0.05$**     | **$\alpha = 0.1$**      | **$\alpha = 0.5$**      | **$\alpha = 1.0$**      |
> | -------------- | ---------------- | ---------------- | ---------------- | ---------------- |
> | FedAvg  | 51.58 ± 2.46     | 51.36 ± 3.82     | 50.60 ± 2.89     | 52.98 ± 0.99     |
> | FedDPD w/o DPPO | 64.47 ± 1.50     | 58.58 ± 3.65     | 55.31 ± 2.35     | 57.67 ± 1.27     |
> | FedDPD         | **64.64 ± 2.23** | **59.96 ± 3.67** | **57.09 ± 3.14** | **58.95 ± 1.37** |
>
> | **Cifar100**   | **$\alpha = 0.1$**      | **$\alpha = 0.5$**      | **$\alpha = 1.0$**      |
> | -------------- | ---------------- | ---------------- | ---------------- |
> | FedFA          | 32.37 ± 3.08     | 28.46 ± 1.61     | 26.21 ± 1.98     |
> | FedDPD         | 37.41 ± 1.12     | 32.44 ± 1.75     | 31.58 ± 1.45     |
> | FedFA + FedDPD | **39.53 ± 1.31** | **34.38 ± 2.13** | **33.19 ± 1.27** |
>
> We will incorporate these additional results into Table 2 in the revised version of the paper.

---

### Author Response · Authors · 2025-12-03
**Summary**

We sincerely thank all reviewers (zK2q, croB, kKa6, Fv9H) and the AC for your careful reading, constructive feedback, and insightful suggestions. We are glad that the reviewers’ concerns have been largely addressed, and we appreciate that reviewer Fv9H would like to raise his score from 4 to 6. Below we summarize the key responses and clarifications provided during the rebuttal.

# Scope & Theorem Clarification

We explicitly clarify that FedDPD is designed for multi-class federated classification, rather than other tasks such as generative modeling or regression. Regarding Theorem 1, we emphasize that it provides an intuition rather than a strict guarantee, and we have discussed the scenarios where its assumptions may not hold. Our experiments show that these issues do not arise in practice, which is also acknowledged by the reviewer zK2q.

# Server-Side Overhead and Scalability

Reviewer kKa6 raised a key concern about the scalability of server workload. We performed a detailed profiling study and solved his concern.

- Under cross-silo settings (10/30/50 clients), we calculated additional time cost and confirmed the overhead remains manageable.
- For extreme settings (100+ clients), we provide a solution: Computationally Efficient FedDPD (shared LoRA) variant, reducing LoRA training from $O(N)$ to $O(1)$, with only slight performance loss while still outperforming all baselines.

# Domain Shift between Stable Diffusion and Client Data

We extended experiments to satellite domain (EuroSAT) and medical domain (ChestMNIST). FedDPD still improves over generative baselines and performs strongly under moderate or high heterogeneity. We provide an deeper analysis of the trade-off between class imbalance reduction and diffusion domain gap, and discuss future extensions, such as using domain-specific generators.

# Additional Baselines & Comparisons

Following reviewer croB's advice, we added more recent SOTA generative FL baselines and showed FedDPD consistently outperforms them. We reproduced Fedssyn, FedDifRC, and DPSDA-FL under our globally imbalanced evaluation setting, and clarified why their performance differs from the original papers (those papers assume globally balanced distributions, while ours intentionally evaluate globally imbalanced settings).

Overall, the reviewers offered insightful and constructive feedback, and we will incorporate these points into the revised version.

---

### Note · Authors · 2026-01-27

I have read and agree with the venue's withdrawal policy on behalf of myself and my co-authors.

---

### Meta-Review · Area_Chair_q1GN · 2026-01-07

**Summary:**

This paper presents a synthetic data generation method on mitigating data heterogeneity in federated learning. This covers both scenarios on imbalanced local datasets as well as imbalanced global data (when we unionize all local datasets). The proposed method is reasonably motivated and demonstrated to improve over a number of baselines.

The reviewers provided relatively borderline scores (6644) originally with multiple concerns. The rebuttal appears substantive and has resolved some concerns which lead to one reviewer (4) updating score. However, despite the interesting idea, I have several concerns:

First, the proposed method appears to miss out on multiple works on federated learning for heterogeneous data situation (both globally and locally skewed datasets) which were published in the recent (2024-2025) NeurIPS/ICML/ICLR cycles. A simple google search on keywords FL and non-IID and imbalanced, long-tailed data should return multiple hits. For example, [*] also uses data generation/augmentation to handle data imbalance.

[*] https://openreview.net/pdf?id=G89r8Mgi5r (NeurIPS-24)

Second, the achieved performance of the proposed method in most cases is too low, especially in CIFAR-100 (<34%) and TinyImageNet (<24%). Even for the CIFAR-10 dataset, the performance is <65% across settings. This level of performance does not quite match state-of-the-art (SOTA) results reported in previous work (on handling federated/skewed data). Some of which does not require data generation. Given this, the impact of data generation in this specific context becomes unclear to me.

I suspect that the overall weak performance in this paper might be caused by the choice of low-complexity pre-trained models. Perhaps choosing a SOTA pre-trained model (such as ViT) could boost performance significantly. The paper should have run ablation studies with different choices of pre-trained model to check if the impact of data generation remains meaningful (rather than becoming marginal) when we use stronger pre-trained model.

In addition, for evaluation under global skewness, there are long-tail variants of CIFAR and ImageNet. For example, this is CIFAR-LT: https://huggingface.co/datasets/tomas-gajarsky/cifar10-lt -- the authors are encouraged to used these datasets for evaluation.

Overall, while the reviewer's opinion of this paper has improved after the rebuttal, there are issues with this paper that were unfortunately not identified during the original reviews. These are nonetheless critical issues that require additional empirical evaluation. The paper also misses out on most recent FL for imbalanced data. Given this, unfortunately, I cannot verify the impact of the current research yet. Practically speaking, the overall weak results across multiple datasets is an indicator that the research is not quite at a mature level for publication (although it did improve over a few selected baselines while ignore several recent ones).

**Reviewer Concerns:**

Reviewer zK2q has outstanding concern. Reviewer Fv9h is satisfied with the rebuttal.
Reviewer croB's review should be disregarded as there were almost no information in the review & there were no interaction.
Reviewer kKa6 agrees most of the concerns have been addressed except for a serious concern on baseline results discrepancy. As explained below, the authors clarified that it was due to being evaluated in a different setting. But, this reveals concerns that the proposed method's performance is too weak compared to other recent SOTA on federated learning (that this work misses) on the balanced global data setting. I have elaborated specifically on this in the main review.

**Reviewer Scores:**

The original scores were 6644. One reviewer who gave a 6 provided little information for the review (despite communication from the previous AC). One reviewer who gave a 4 appeared satisfied with the rebuttal. I assume his/her score will be raised to 6. The other two reviewers have sustained concerns. One (who gave a 6) think this method still does not address sufficiently the data heterogeneity problem and suggests other settings where the approach will likely not work. Another (who gave a 4) questions the discrepancy of performance reported in the previous work. The authors have provided a sensible answer, highlighting that the experiment settings here are different (where the evaluation is centered on global skewness). The rebuttal also provided additional experiments regarding settings where local datasets are skewed but global data are balanced. I am not sure whether the reviewer will find this result convincing (if he/she was able to participate fully in the discussion) but I do not. These results on CIFAR-10 (when global dataset is balance) are far below the state-of-the-art performance.

---

### Decision · Program_Chairs · 2026-01-26

Reject